# Serial processing of kinematic signals by cerebellar circuitry during voluntary whisking

Susu Chen[1], George J. Augustine[2,3] & Paul Chadderton [1]

Purkinje cells (PCs) in Crus 1 represent whisker movement via linear changes in firing rate, but the circuit mechanisms underlying this coding scheme are unknown. Here we examine the role of upstream inputs to PCs—excitatory granule cells (GCs) and inhibitory molecular layer interneurons—in processing of whisking signals. Patch clamp recordings in GCs reveal that movement is accompanied by changes in mossy fibre input rate that drive membrane potential depolarisation and high-frequency bursting activity at preferred whisker angles. Although individual GCs are narrowly tuned, GC populations provide linear excitatory drive across a wide range of movement. Molecular layer interneurons exhibit bidirectional firing rate changes during whisking, similar to PCs. Together, GC populations provide downstream PCs with linear representations of volitional movement, while inhibitory networks invert these signals. The exquisite sensitivity of neurons at each processing stage enables faithful propagation of kinematic representations through the cerebellum.

---

[1] Department of Bioengineering and Centre for Neurotechnology, Imperial College London, London SW7 2AZ, UK. [2] Lee Kong Chian School of Medicine, Nanyang Technological University, 11 Mandalay Road, Singapore 308232, Singapore. [3] Institute of Molecular and Cell Biology, 61 Biopolis Drive, Proteos, Singapore 138673, Singapore. Correspondence and requests for materials should be addressed to P.C. (email: p.chadderton@imperial.ac.uk)

Animals actively probe and interact with the world by moving to acquire sensory information. Self-motion has sensory consequences that enable the nervous system to guide and adjust future movement, with sensorimotor brain circuits constantly engaged to optimise this process[1]. For tactile sensation, rodents rhythmically sweep their whiskers back and forth to scan the proximal surrounding. Such active whisking enables animals to explore, identify and discriminate objects with impressive degrees of sensitivity and capability[2]. This behaviour has served as a well-defined paradigm to study active sensory processing and has yielded many insights into the neuronal circuit basis of sensorimotor control[3-10]. The cerebellum is strongly implicated in sensorimotor processing[11], and recent studies in the vibrissae regions of the rodent cerebellum have highlighted its functional role in the control of voluntary whisker movement[8] and in sensorimotor learning tasks[12]. However, the precise role(s) performed by this structure during voluntary whisking is poorly understood and the implications for cerebellar function remain unclear.

The organisation of the cerebellar cortex is relatively simple and is comprised of a densely packed input layer, the granule cell layer (GCL), which provides excitatory drive via parallel fibres (PFs) to Purkinje cell (PC) dendrites and molecular layer interneurons (MLIs). PCs integrate excitatory and inhibitory synaptic inputs from PFs and MLIs in order to shape spike output for the entire cerebellar cortex[13-16]. In lobule Crus 1, the majority of PCs encode whisker set point through linear bidirectional changes in simple spike firing rate[17]. Such remarkable linear encoding of a single kinematic parameter requires precise integration of both excitatory (PF) and inhibitory (MLI) inputs that together provide whisking-related signals to the dendrites of PCs. However, the functional contribution of PFs and MLIs to the generation of PC movement signals is not known.

Because granule cells (GCs) transform mossy fibre (MF) input into excitatory PF drive to both MLIs and PCs, it is essential to determine how these cells encode whisker movement prior to processing at subsequent stages of the cerebellar circuit. GCs are the smallest and most abundant neurons in the brain. They receive only a small number of MF inputs (<7), suggesting that GCs may individually encode movement more selectively than PCs. However, their small size and high packing density has precluded measurement of their activity during whisking.

GABAergic inhibitory interneurons (INs) substantially influence information transmission at multiple stages in the cerebellar cortex. Golgi cells (GoCs) provide feedforward and feedback inhibition that mediates the excitability and gain of GCs in the input layer[18-20], whereas MLIs exert potent feedforward and lateral inhibition to regulate the firing rate and spatiotemporal dynamics of PC simple spiking[14-16, 21-23]. As a result of this organisation, close examination of the inhibitory network is also required to obtain a complete understanding of information flow through the cerebellar cortex.

In this study, we reveal the circuit mechanisms that govern bidirectional linear encoding of whisker movement and establish how information about whisker motion is conveyed by excitatory and inhibitory inputs to PCs. We have used patch clamp recordings of the activity of GCs to investigate the representation of movement in these cells. Approximately one-third of PCs exhibit spike rate reductions during free whisking[17]. Unlike downstream PCs, whisker movement is associated solely with increased GC activity in the cerebellar input layer. Patch clamp recordings of the activity of INs reveal that reciprocal firing patterns of PCs are generated via the di-synaptic GC-IN-PC inhibitory pathway[24]. Our results demonstrate that processing of whisker movement signals occurs sequentially at successive stages of the cerebellar circuit in order to generate precise bidirectional estimates of whisker position in PCs.

## Results

### Widespread depolarisation within GC layer during whisking.
Around 1 out of 3 of movement-responsive PCs exhibit reduced firing rates during free whisking[17], but the origin of these decreases in activity is unclear. A reduction in net excitatory drive to PCs could result from reduced activity within upstream populations of MFs and PFs, or alternatively via inhibitory operations within the cerebellum. To examine the underlying mechanism, we examined GC and IN activity during epochs of voluntary whisking.

We performed whole-cell (WC; $n = 32$, $N = 22$ mice) and cell-attached (CA; $n = 13$, $N = 11$ mice) patch clamp recordings from GCs in the vibrissal areas of the cerebellar cortex (lobule Crus 1) of awake mice. To correlate GC activity with whisker movement, we simultaneously tracked spontaneous whisking via a high-speed camera (Fig. 1a, b). In the WC configuration, GCs were readily identified by their characteristic in vivo electrophysiological properties[25-28], including high input resistance ($0.63 \pm 0.07$ GΩ, $n = 32$) and fast membrane time constant ($\tau = 5.9 \pm 0.4$ ms, $n = 32$). These measurements are in good agreement with the data obtained from the lateral hemispheres of anaesthetised rodents[25, 29, 30]. The mean resting membrane potential of GCs was $-63.8 \pm 1.0$ mV ($n = 32$). In both WC and CA recordings, GCs exhibited low baseline firing rates (Fig. 1b, c; WC: $0.8 \pm 0.5$ Hz, $n = 32$; CA: $4.8 \pm 1.3$ Hz, $n = 13$), bursting patterns of spike output[31] as reflected in high coefficient of variation of interspike interval (CV of ISI in WC: $4.9 \pm 1.6$, $n = 12$; CV of ISI in CA: $2.9 \pm 0.5$, $n = 13$; $P = 0.3$, Mann–Whitney $U$ test), and short half-width of action potentials ($0.31 \pm 0.11$ ms, $n = 13$ in CA). Although the mean firing rate of CA recordings was significantly different from that of WC recording ($P < 0.001$, Mann–Whitney $U$ test), this was likely due to our inability to identify silent GCs in CA mode. Accordingly, the whole-cell data revealed a substantial fraction of GCs ($n = 26/32$) that remained silent in the absence of whisker movement (Fig. 1c).

A total of 26 out of 32 GCs exhibited significant differences in membrane potential between quiet and whisking periods ($P < 0.05$, Kolmogorov–Smirnov test assessed on a cell-by-cell basis; Fig. 1d). In these cells, whisking was associated with significant membrane depolarisation ($-61.5 \pm 1.0$ mV to $-58.9 \pm 1.1$ mV; $n = 26$; $P < 0.001$, Wilcoxon signed-rank test; Fig. 1e), indicating a widespread increase in excitatory drive to the GCL. Unlike downstream PCs, reductions in GC activity during movement were never observed. To address the precise temporal relationship between membrane potential depolarisation and whisking, we computed normalised mean-subtracted cross correlations between whisker position and corresponding voltage traces, centred on the onset of individual whisking epochs and averaged across trials (Fig. 1f). In the majority of GCs ($n = 26/32$), we observed significant positive correlations between whisker position and membrane potential (Fig. 1f, g). On average, depolarisation peaked at $9.3 \pm 4.8$ ms ($n = 26$) prior to the onset of whisking (Fig. 1g, Supplementary Fig. 1), but approximately one-third ($n = 8/26$) of GCs lagged movement, and overall there was a broad distribution of latencies between GC depolarisation and movement (Fig. 1g). Neuronal activity that precedes movement is consistent with observations in PCs, where changes in simple spiking typically precede whisking onset[17], and is consistent with a movement signal that is internally, rather than externally, generated[28, 32].

### Encoding of movement via MF input to GCs.
Several factors could contribute to the observed depolarisation of membrane potential during free whisking: (1) enhanced frequency of MF input[27, 28, 33], as revealed by an increase in the rate of excitatory postsynaptic currents (EPSCs); (2) increased amplitude of

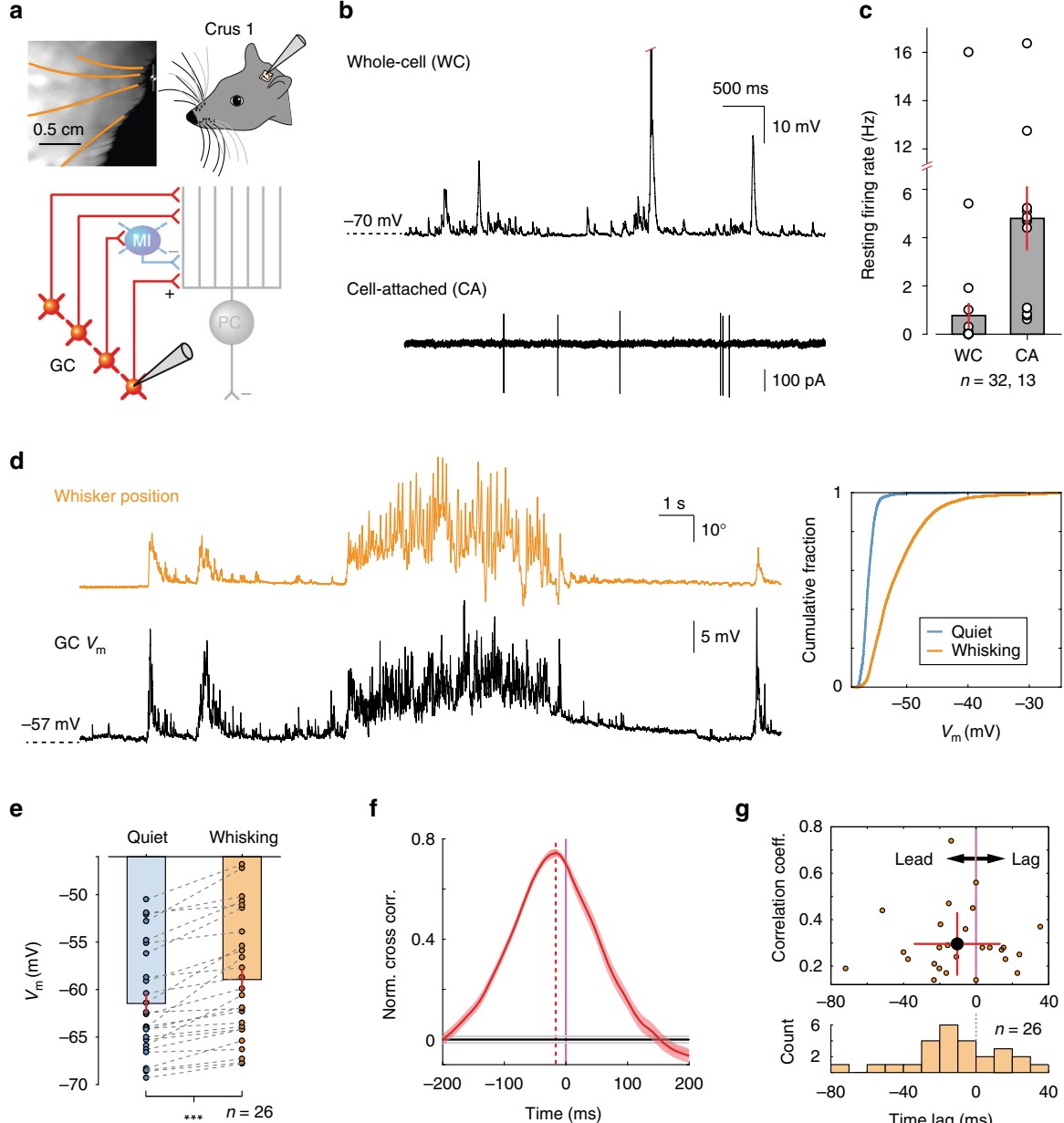

**Fig. 1** Whisking drives membrane potential depolarisation in cerebellar granule cells of awake mice. **a** *Top left*: videography of a head-restrained mouse with four traced whiskers (from row C, labelled in *orange*). *Top right*: schematic of experimental setup. Patch clamp recordings were made from lobule Crus 1 of the cerebellar cortex. *Bottom*: schematic representation of cerebellar circuit highlighting granule cells in the input layer. *GC* granule cell, *MI* molecular layer interneuron, *PC* Purkinje cell. **b** *Top*: whole-cell (*WC*) patch clamp recording from GC. *Red tick* indicates truncated action potential. *Bottom*: cell-attached (*CA*) recording from putative GC. **c** Resting GC firing rates in the absence of movement ($n = 32$ WC, $n = 13$ CA). A total of 26 out of 32 GCs were silent in WC configuration. **d** *Left*: example traces of whisker position (*orange*; upward deflection indicating protraction) and simultaneously recorded GC membrane potential $V_m$ (*black*). *Right*: cumulative probability of $V_m$ distribution during quiet and whisking epochs for this GC ($P < 0.001$, Kolmogorov–Smirnov test). **e** Mean $V_m$ in periods of quiet and whisking for all GCs that showed statistically significant differences in $V_m$ distribution between two conditions. $V_m$ is significantly depolarised during whisking (***$P < 0.001$, Wilcoxon signed-rank test, $n = 26$). **f** Normalised cross-correlation between whisker position and $V_m$ for a single GC. *Purple line* at zero depicts whisking onset. *Vertical red dashed line* highlights the peak in the cross correlogram, indicating temporal relationship between behaviour and GC $V_m$. *Horizontal grey lines*: 95% confidence interval. GC exhibited depolarisation preceding the onset of whisking. **g** *Top*: scatter plot, indicating temporal relationship and strength of correlation between whisker position and GC $V_m$. *Black-filled circle*: Mean ± S.D. *Bottom*: histogram of temporal relationship between behaviour and GC $V_m$ ($n = 26$)

individual EPSC events and/or (3) elevated sensitivity to synaptic input mediated by enhanced glutamate spillover from neighbouring synapses[28, 29, 34]. To understand the relationship between GC depolarisation and whisker movement, we performed voltage-clamp recordings to measure EPSCs[25]. GCs were clamped at −70 mV (close to the reversal potential for synaptic inhibition) and EPSCs were recorded during bouts of quiescence and whisking ($n = 11$ cells, $N = 7$ mice, Fig. 2a). Whisking bouts were associated with an overall increase in inward current (Fig. 2b). Analysis of individual EPSC waveforms (Supplementary Fig. 2) revealed that both the amplitude and time-course of individual MF inputs were unchanged between

quiescence and whisking ($17.5 \pm 2.4$ pA to $18.1 \pm 2.4$ pA, $P = 0.45$ for EPSC amplitude; $0.18 \pm 0.02$ to $0.19 \pm 0.03$ ms, $P = 0.25$ for 20–80% rise time, Wilcoxon signed-rank test, $n = 11$, Fig. 2c, d, f, g). In contrast, whisking was associated with a significant increase in EPSC rate ($11.7 \pm 2.0$ to $24.7 \pm 6.0$ Hz; $P < 0.001$, Wilcoxon signed-rank test, $n = 11$; Fig. 2e, h). These results demonstrate that movement-related increases in inward current, and resulting membrane potential depolarisation, are predominately caused by elevated rates of MF synaptic input during movement.

**MF input rate determines GC kinematic tuning**. Having established that movement is principally associated with increases in EPSC rate, we explored whether these excitatory synaptic inputs represented specific kinematic parameters of whisking[17]. We examined the relationship between EPSC rate and: (1) raw whisker position, (2) the slowly varying parameter, set point (Fig. 3a) and (3) the rapidly varying parameter, phase (Supplementary Fig. 3a). In 11 recordings, the majority of GCs showed selectivity to whisker position and set point ($n = 6$ for

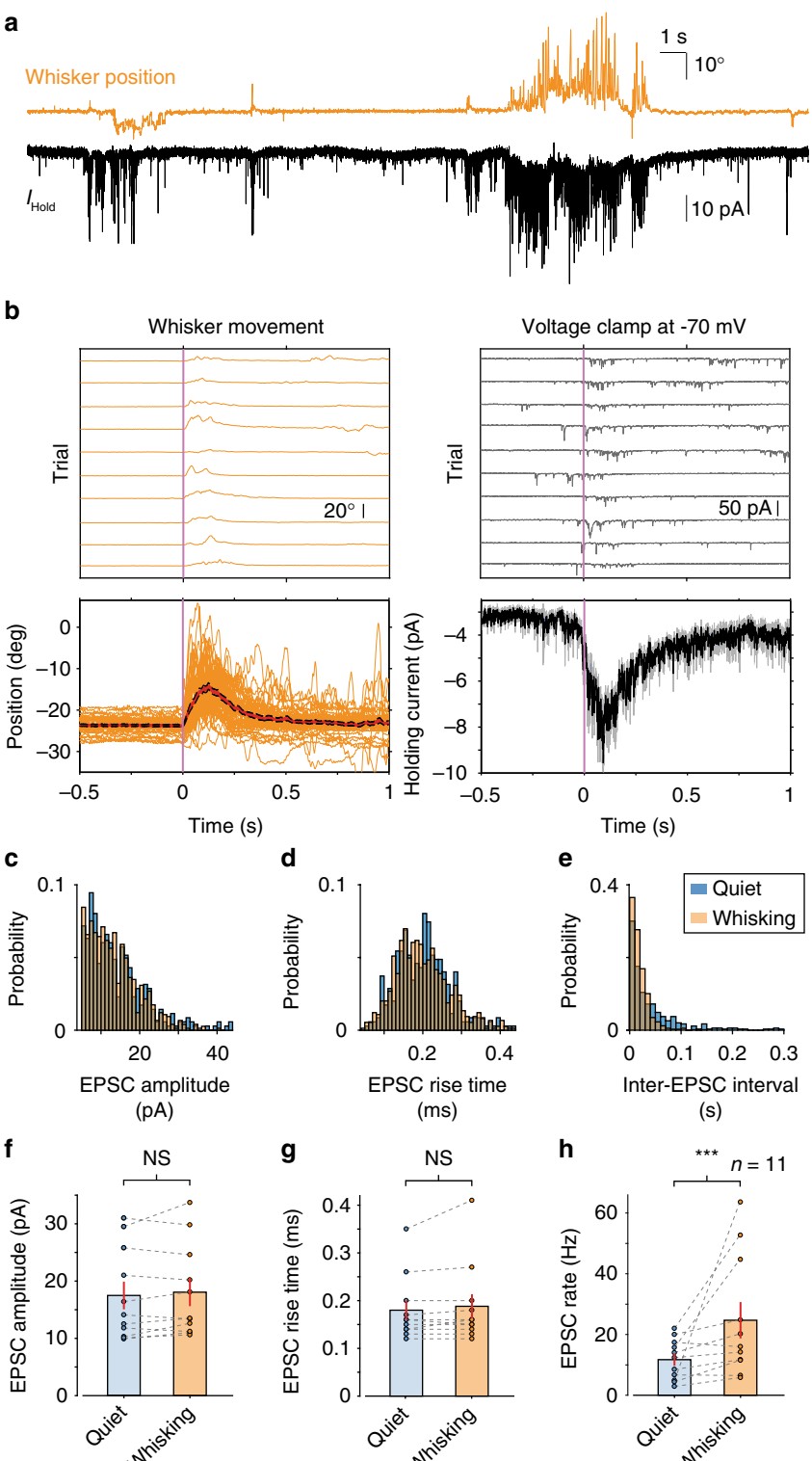

position, $n = 7$ for set point, $P < 0.05$, Kolmogorov–Smirnov test comparing distribution of angle at all times to its distribution at times of EPSC occurrence for each recording; Fig. 3b). Tuned GCs exhibited 'preferred angles' for position and set point, which corresponded to the highest rates of MF input. Individual GC tuning curves remained relatively broad such that progressive deviations from the preferred angle were associated with progressive, but gradual, reductions in EPSC rate. A small number of GCs also exhibited significant tuning to whisking phase ($n = 2$ out of 11, $P < 0.05$, Kuiper's test comparing distribution of phase at all times to its distribution at times of EPSC occurrence for each recording; Supplementary Fig. 3b). These results demonstrate that GC sensitivity to both slow and fast kinematic parameters is conferred via changes in the rate of MF input. Similarly, we compared the relationship between membrane potential and features of movement in current-clamp recordings. GCs showed selectivity to whisker position and set point demonstrating broad angular tuning (Fig. 3c). Across the population, the voltage range between 'preferred' and 'least-preferred' angle (see 'Methods' section) was $10.6 \pm 1.8$ mV for position and $9.0 \pm 1.4$ mV for set point ($n = 26$, range: 1.1–30.0 mV for position, 1.5–27.4 mV for set point; Fig. 3d), demonstrating that changes in whisker angle alone are associated with substantial depolarisations in individual GCs.

We next examined the relationship between EPSC rate tuning and membrane potential directly. In a small number of GCs ($n = 5$), we were able to perform consecutive voltage- and current-clamp recordings, and directly compare movement-related changes in EPSC rate and membrane potential. We observed a close correspondence in both the profile (as in Fig. 3b, c), and preferred angle (Fig. 3e) of input (EPSC) tuning and membrane potential modulation, indicating that the tuning of MF input accounts for the subthreshold selectivity of GCs during whisking.

**Sharp and selective output amongst GC populations.** To determine how an increased rate of MF EPSCs influences GC output during whisking, we measured GC firing patterns (Fig. 4a). In approximately a third of GCs, whisker movements were associated with enhanced spiking (Fig. 4b, c). Overall, the firing rate in WC and CA recordings increased from $1.9 \pm 0.6$ to $4.2 \pm 1.1$ Hz ($P < 0.001$, Wilcoxon signed-rank test, $n = 45$ WC and CA recordings; Fig. 4d). In GCs with enhanced spike output, action potentials occurred in bouts of high-frequency bursts (Fig. 4a, Supplementary Fig. 1) with a mean instantaneous firing frequency of $219 \pm 56$ Hz in individual bursts (defined as groups of spikes with ISI less than 50 ms[28]). Unlike brief sensory-evoked bursting in anaesthetised rodents[25, 30], movement-related bursts were longer lasting, containing on average $31.5 \pm 17.9$ spikes with an inter-burst interval of $2.1 \pm 0.8$

s. Accordingly, the mean CV of ISI was extremely high for GCs ($3.6 \pm 0.8$, $n = 25$).

We next compared membrane-potential- and action-potential- (i.e., spiking output) tuning to kinematic features in WC recordings (Fig. 4e–h). GC output was tuned to both position and set point ($n = 9$ out of 12 cells; $P < 0.05$, Kolmogorov–Smirnov test comparing distribution of angle at all times to its distribution at times of action potential occurrence for each recording), and also phase ($n = 2$ out of 12 cells; $P < 0.05$, Kuiper's test comparing distribution of phase at all times to its distribution at times of action potential occurrence for each recording; Supplementary Fig. 3c). In all cases, the preferred angle/phase corresponded tightly between membrane potential and spike output (Fig. 4e–i, Supplementary Fig. 3b). However, GC output tuning was considerably more selective than input tuning as indicated by the modulation depth for EPSC and spike rate between 'preferred' and 'least preferred' angles (Fig. 4j; see 'Methods' section). Overall, our results demonstrate that GCs are highly selective for distinct kinematic features of whisking. This selectivity is conferred predominately via increases in MF EPSC rate, which causes predictable membrane potential depolarisations and action potential firing at preferred angles.

To determine how GC populations respond during changes in whisker set point, the tuning curves of individual GCs were normalised with respect to the range of movement observed during individual recordings (see 'Methods' section). This enabled us to average across cells for a given set point percentile and obtain the mean tuning curve for the population as a whole. Whereas individual GCs displayed sharp tuning at one or multiple set point positions (Fig. 5a), the mean tuning function displayed a nearly monotonic dependence upon whisker set point change from resting position, with increasing activity in the direction of protraction or retraction (Fig. 5b). Regression analysis performed, respectively, over the range of retraction and protraction from resting position revealed significant linear relationships between population firing rate and relative set point change (retraction: $R^2 = 0.32$, protraction: $R^2 = 0.64$, $P < 0.05$, ANOVA, Fig. 5b). Therefore, populations of GCs with distinct and relatively selective tuning properties transmit set point information to downstream targets (PCs and INs) via elevated firing rates.

**Bidirectional IN firing rate change during whisking.** PCs in Crus 1 represent whisker movements via both increases and decreases in simple spike activity. However, upstream GC activity is always enhanced during whisking, suggesting that downstream inhibition plays a key role in determining the sign and degree of bidirectional PC simple spiking. To examine the role of INs during behaviour, we performed patch clamp recordings from INs in this lobule (Fig. 6a). INs were distinguished on the basis of their resting firing rates and the absence of complex spiking

**Fig. 2** Elevated rates of mossy fibre synaptic input during whisking bouts. **a** Whisker position (*orange*; upward deflection indicating protraction) and simultaneously recorded holding current from GC-voltage clamped at −70 mV (*black*; downward deflections correspond to inward currents). **b** *Top left*: whisker position from 10 representative whisking epochs (trials). *Top right*: corresponding GC holding current. Voltage-clamp sweeps have been smoothed for visual clarity. *Bottom left*: raw position traces (*orange*) and mean whisker position (*red line*; *black*: S.E.M.) for all trials ($n = 50$ trials). *Bottom right*: mean holding current (*black line*; *grey*: S.E.M.) for all trials ($n = 50$ trials). **c** Normalised histogram of EPSC amplitude for a single GC during quiet and whisking epochs. No significant change was observed between the two conditions ($P = 0.77$, Kolmogorov–Smirnov test). **d** Normalised histogram of EPSC 20–80% rise-time for a single GC during quiet and whisking epochs. No significant change was observed between the two conditions ($P = 0.07$, Kolmogorov–Smirnov test). **e** Normalised histogram of EPSC inter-current interval for a single GC during quiet and whisking epochs. The two distributions were significantly different ($P < 0.001$, Kolmogorov–Smirnov test), with a larger proportion of short intervals during whisking. **f** Mean EPSC amplitude in periods of quiet and whisking for all GCs. No significant change was observed between the two conditions ($P = 0.45$, Wilcoxon signed-rank test, $n = 11$). **g** Mean EPSC 20–80% rise-time in periods of quiet and whisking for all GCs. No significant change was observed between the two conditions ($P = 0.25$, Wilcoxon signed-rank test, $n = 11$). **h** Mean EPSC rate in periods of quiet and whisking for all GCs. EPSC rate was significantly elevated during whisking epochs (***$P < 0.001$, Wilcoxon signed-rank test, $n = 11$)

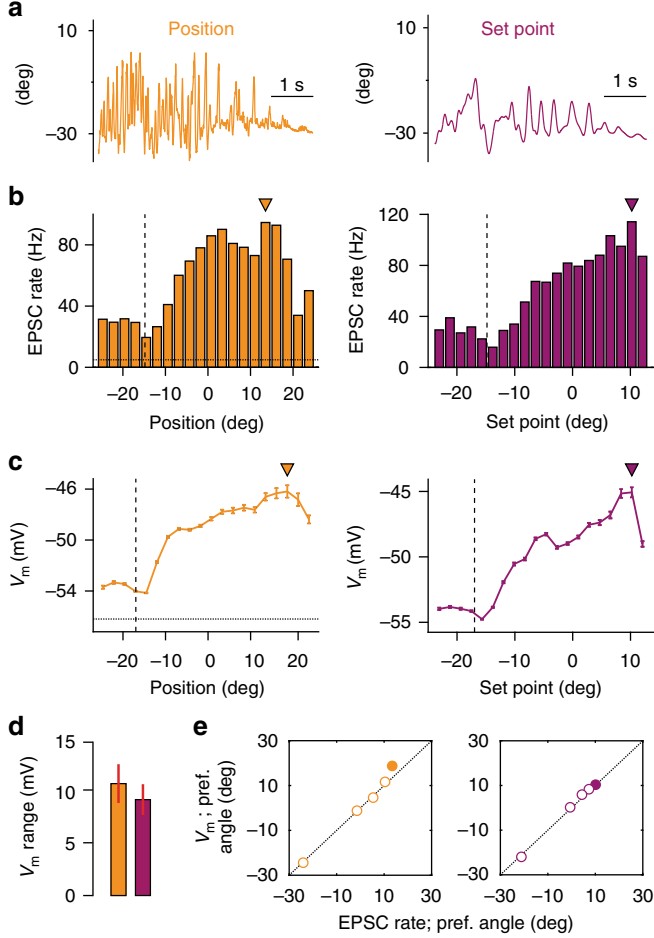

**Fig. 3** Kinematic tuning of granule cell synaptic input and membrane potential. **a** Snapshot of fluctuating whisker position (*orange, left*) during movement and corresponding change in whisker set point (*purple, right*). **b**. Histogram of EPSC rate with respect to whisker position (*left*) and set point (*right*) for a single GC. *Filled triangles*: preferred angles. *Dashed horizontal line* indicates average quiescent EPSC rate. *Vertical line* indicates whisker resting position. **c** Average $V_m$ with respect to whisker position (*left*) and set point (*right*) for the same GC. The overall profile of angular tuning is similar to **b**. *Filled triangles*: preferred angles. *Dashed horizontal line* indicates average quiescent $V_m$. *Vertical line* indicates whisker resting position. **d** Dynamic range of $V_m$ tuning with respect to whisker angle ($n = 26$). Value represents average of difference between $V_m$ preferred and least-preferred angle for individual GCs. **e** Correspondence between preferred angle for EPSC rate (measured in voltage-clamp mode) and $V_m$ (measured in current-clamp mode) for all GCs ($n = 5$). *Filled circle*: GC shown in **b**, **c**

during recordings in both WC and CA[35] configurations (Fig. 6b, Supplementary Fig. 4).

When mice were not whisking, INs fired tonically (whole-cell: $20.0 \pm 2.6$ Hz, $n = 5$, and cell-attached: $33.1 \pm 4.7$ Hz, $n = 40$, $N = 32$ mice, Fig. 6b), in agreement with previous studies[36–39]. During whisking, a large fraction of INs exhibited significant changes in firing rate ($P < 0.05$, Wilcoxon signed-rank test, $n = 35$ out of 45; Fig. 6c). Approximately two-thirds of such INs increased their spiking during whisking ($n = 23$ out of 35, Fig. 6d), while activity decreased in the remaining third ($n = 12$ out of 35, Fig. 6e). Changes in INs firing rate were non-uniform, regarding both the sign and magnitude of modulation across the population (as large $+ 458\%$ and $-66\%$ changes in firing rate during whisking epochs). INs with low baseline firing rate during

non-whisking epochs did not show any bias towards increasing their spiking during whisking or vice versa for cells with high baseline rate, indicating that the direction and magnitude of rate change were independent of the cell's spontaneous firing rate (Fig. 6c).

**Linear relationship between firing rate and whisker position.** PCs discharge tonically at rest, and by integrating excitatory inputs from GCs and inhibitory inputs from MLIs, form a linear neural code to represent voluntary whisking[17]. The same could be true for INs[40], which also display ongoing firing activity and share similar synaptic inputs with PCs (both are innervated by GCs and other INs). To test this prediction, we determined the relationship between IN instantaneous firing rate and whisker position (Fig. 6d, e; see 'Methods' section). Strong linear relationships were revealed in nearly half of INs ($n = 19$ out of 43, linear regressions fit: $R^2 = 0.96 \pm 0.01$; $P < 0.05$, ANOVA, $n = 19$; Supplementary Table 1) in a directionally selective manner.

Two types of linear encoding schemes were present in INs: unidirectional ($n = 8$, Fig. 7a) and bidirectional ($n = 11$, Fig. 7b) with respect to whisker angle. Unidirectional INs (Fig. 7a) showed piecewise linear correlation within a range of whisker positions corresponding to either forward or backward movements (relative to resting position). In contrast, bidirectional INs (Fig. 7b) responded during both forward and backward whisker movement and were capable of continuously representing whisking by firing rate change. In these cells, whisker angles on one side (i.e., either protracted- or retracted-) were associated with firing rate increases proportional to positional change from resting point, whereas, movement on the opposite side was associated with linear reductions in firing rate (Fig. 7b). To provide a full picture of such linear representation across the population of INs, each IN's spiking was normalised with respect to its baseline rate and the relative change in whisker position was obtained by subtracting the corresponding resting position. Both types of INs demonstrated almost perfect linear relationships (unidirectional: $R^2 = 0.98 \pm 0.01$, $n = 8$; bidirectional: $R^2 = 0.94 \pm 0.01$, $n = 11$; $P < 0.05$, ANOVA) between relative changes in firing rate and mean whisker position over a certain range (Fig. 7c). The average gain of INs, which was defined as the absolute value of the slope of each linear regression fit, was $25.0 \pm 4.6$ Hz/degree ($n = 19$; bidirectional: $31.4 \pm 7.1$ Hz/degree, $n = 11$; unidirectional: $16.2 \pm 3.8$ Hz/degree, $n = 8$; $P = 0.12$, Mann–Whitney $U$ test, Fig. 7d). In comparison to PCs, INs exhibited larger gain values across the population (IN: $25.0 \pm 4.6$ Hz/degree, $n = 19$; PC: $15.8 \pm 2.1$ Hz/degree, $n = 44$; $P < 0.001$, Mann–Whitney $U$ test), meaning INs exhibited larger changes in firing rate in order to encode the same degree of movement, and indicating that these cells may be more susceptible to rate saturation (i.e., boundary effects). To test the fidelity of movement encoding by INs, transfer functions were computed from individual spike trains and corresponding whisker positions ($n = 10$, recordings with a correlation coefficient between whisker position and instantaneous firing rate $> 0.25$). Using this approach, it was partially possible to predict the dynamics of whisking trajectory in real time (Supplementary Fig. 5a). The reconstruction from single INs captured the dynamics of the slowly varying whisking kinematic parameter, set point, although the amplitudes of reconstructed trajectories were attenuated, suggesting a low-pass filtered representation of movement. Overall, INs were modest predictors of whisking trajectory with an average correlation coefficient value of $0.42 \pm 0.08$ (range: 0.14–0.76, $P < 0.01$, $n = 10$; Supplementary Fig. 5b) between reconstruction and real set point. Taken together, our results confirm that selective encoding of

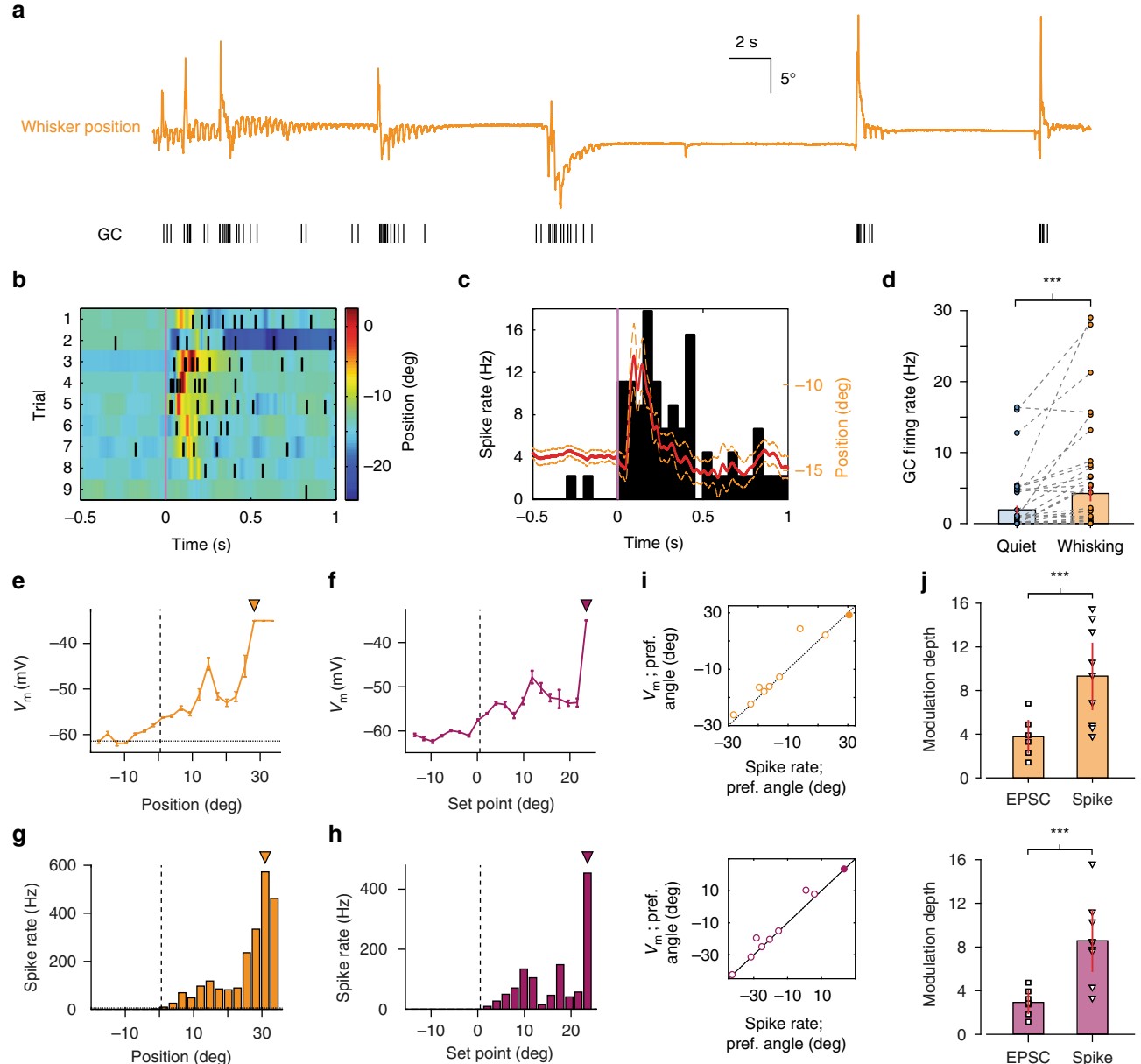

**Fig. 4** Kinematic tuning of granule cell output. **a** Whisker position (*orange*; upward deflection indicating protraction) and corresponding GC spike train. **b** *Colour-coded* whisker movement from nine consecutive epochs (trials) and corresponding spike raster from single GC. **c** Mean whisker position (*red line*; *orange*: S.E.M.) and peri-event time histogram (*PETH*) of GC spiking for all trials. Firing rates were computed from PETH with 50 ms bin size. **d** GC firing rate changes between quiet and whisking epochs from WC and CA recordings (***$P < 0.001$, Wilcoxon signed-rank test, $n = 32$ WC and $n = 13$ CA). **e**, **f** Average $V_m$ with respect to whisker position **e** and set point **f** for a single GC. *Filled triangles*: preferred angles. *Dashed horizontal line* in **e** indicates average quiescent $V_m$. *Vertical line* indicates whisker resting position. **g**, **h** Histogram of spike rate with respect to whisker position **g** and set point **h** for the same GC. The overall profile of angular tuning is similar to $V_m$, but sharper. *Filled triangles*: preferred angles. *Dashed horizontal line* in **g** indicates resting firing rate. *Vertical line* indicates whisker resting position. **i** Correspondence between preferred angle for $V_m$ and spike rate for all GCs ($n = 9$). *Filled circle*: GC shown in **e–h**. **j**. Modulation depth of EPSC rate (i.e. synaptic input) and GC spike rate (i.e., output) with respect to whisker position (*top*) and set point (*bottom*). Modulation depth was significantly higher for output (***$P < 0.001$, Mann–Whitney $U$-test; $n = 6$ input and $n = 9$ output for position; $n = 7$ input and $n = 9$ output for set point)

kinematic features by GCs, and bidirectional modulation of synaptic inhibition are both necessary to account for the physiological patterns of PC simple spiking activity during voluntary whisking.

## Discussion
Understanding the neural representation of motor behaviour requires detailed examination on how single neurons encode movement. Here we have exploited a well-defined model, the mouse whisker system, to probe how single neurons in the cerebellar cortex encode patterns of self-generated movement. We provide the first patch clamp recordings from cerebellar GCs and INs in lobule Crus 1 of awake mice, and demonstrate the sensitivity of these cells to whisker movement. GCs receive excitatory MF synaptic input that represents both fast and slow kinematic features of whisking. While a relatively small fraction of GCs

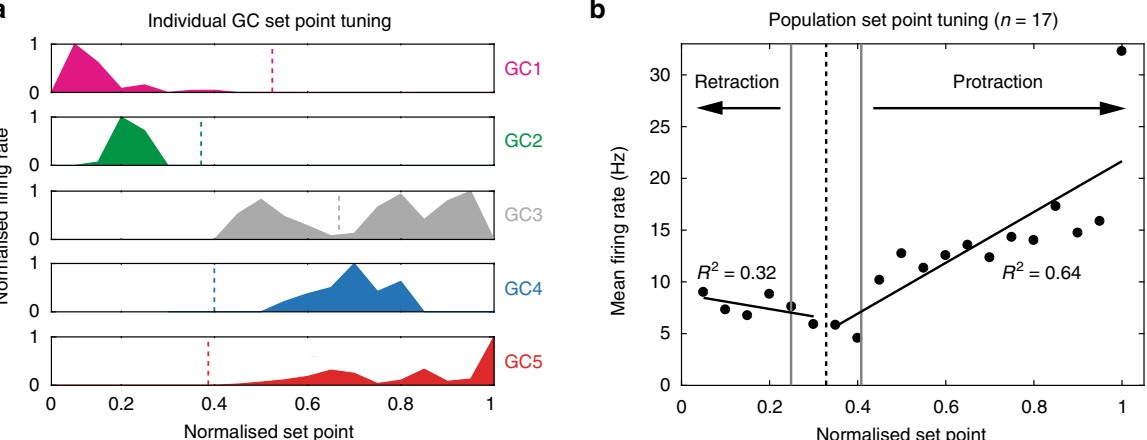

**Fig. 5** Set point tuning of cerebellar granule cells. **a** Five GC examples (GC1–GC5; WC data) demonstrating significant tuning of spike output to whisker set point ($P < 0.05$, Kolmogorov–Smirnov test). Firing rate was normalised with respect to the maximal rate in each tuning curve. Set point is normalised as percentile for comparison among cells. Individual GCs demonstrated sharp peaks at one or multiple set points. *Vertical dashed lines* indicate corresponding mean resting positions (when whiskers are not moving) for each GC. **b** Mean tuning curve for all GCs ($n = 17$) exhibiting significant rate modulation by set point. The data are represented in percentile across population of the whisker data with distinct ranges of set point variable. *Black vertical dashed line* indicates the mean resting position and *grey dotted lines* mark S.E.M. Linear regressions were performed over the range of retraction ($R^2 = 0.32$) and protraction ($R^2 = 0.64$), respectively

show tuning to whisking phase, the vast majority of GCs receive information about set point, which is conveyed to downstream neurons via high-frequency burst firing in a subset of neurons. Despite receiving increased excitatory drive via PFs, a significant fraction of interneurons in the molecular layer display firing rate reductions during free whisking, in a manner similar to PCs[17]. Our results indicate that whisker signals are subject to serial processing within the cerebellar cortex in order to accurately represent movement (Fig. 8): within the GCL layer, whisker-related MF input is integrated within populations of GCs, generating linear increases in PF activity that encode whisker position. Downstream within the molecular layer, broad integration of excitatory PF input and sign reversal via local inhibition[24] occur to implement robust linear bidirectional representations of whisker position in both other INs and PCs.

GCs are small in size and vast in number, providing the sole source of excitatory drive within the cerebellar cortex, but little is known about the properties of these neurons in awake, behaving animals. In the absence of movement, GCs display low baseline firing rates (Fig. 1c). During whisking, the vast majority of GCs undergo membrane potential depolarisation due to increased rates of MF input. Notably, the amplitude and time-course of individual EPSCs is unchanged between quiet and whisking periods. This suggests that the identity of active MF inputs may not change between bouts of quiescence and whisking (unlike, e.g., GCs in the flocculus[27]), and mechanisms such as short-term plasticity (e.g., synaptic depression) do not influence the overall profile of EPSC amplitude between these two conditions. The observed increase in the rate of synaptic input during whisking resembles dramatic increases in excitability observed in lobule V GCs when mice are running[28, 41]. However, individual GCs in Crus 1 receive rate-modulated input that confers kinematic tuning to specific features of whisking behaviour.

A large proportion of GCs exhibit tuning to positional changes, in particular with respect to the slowly varying kinematic parameter, set point, but also to whisker phase. The rate of MF input (EPSCs) varies with respect to whisker angle/phase, and we observed a direct correspondence between the tuning of MF input rate, membrane potential and action potential output in individual GCs (Figs. 3e and 4i). In addition to excitatory MF input, GCs receive Golgi cell inhibition that can control GC output. Although

we have not measured Golgi cell inhibition directly, our results indicate that this source of inhibition does not alter the profile of kinematic tuning of individual GCs. Golgi cell inhibition remains likely to play an important role in regulating the overall excitability of the GCL[19, 30], and the timing of GC output[18], and further work is required to confirm these proposals. However, our results show that MF input rate governs the subthreshold tuning of individual GCs during voluntary whisker movement.

GC membrane potential tuning is typically rather broad (Figs. 3c and 4e, f), but the requirement for substantial depolarisation to reach action potential threshold ensures that individual GC output is selective, and GCs only fire within a limited range of whisker angles (Figs. 4g, h and 5a). The heterogeneous tuning of different GCs ensures that, while individual cells remain selective, whisker movements are associated with monotonic increases in excitability in the GC output at the population level (Fig. 5b).

These measurements from the input layer support the notion that whisker input to the cerebellar cortex is dense and widespread[8, 17, 42, 43], and that efferent rather than re-afferent drive dominates cerebellar activity during free whisking (Fig. 1g, Supplementary Fig. 1). Classical cerebellar theory predicts that the input layer of the cerebellar cortex employs a sparse coding principle in GC populations, in which only a small fraction of GCs generates output in response to a given MF input. Such a scheme could facilitate the information storage capacity of PCs by allowing the discrimination of the greatest amount of distinct PF input patterns[44–46]. Here we observed that a large fraction of GCs (approximately one-third) are activated during whisking, implying a shift from sparse to a dense mode of activation during such behaviour[28, 32]. However, the selectivity of individual GCs to whisker position may ensure that overall low firing rates are preserved on a moment-by-moment basis. Measures of population sparseness will ultimately require simultaneous monitoring of many GCs[32, 47] during whisker movement (or monitoring of PF activity[48]) to examine the fraction and identification of active GCs in Crus 1.

Recent studies have revealed the convergence of functionally distinct afferent inputs onto single GCs across the mouse cerebellum[8, 49–51]. The congruent arrangement of somatosensory and motor input to lobule Crus 1 means that individual GCs might integrate convergent sensory- and motor-related signals about the

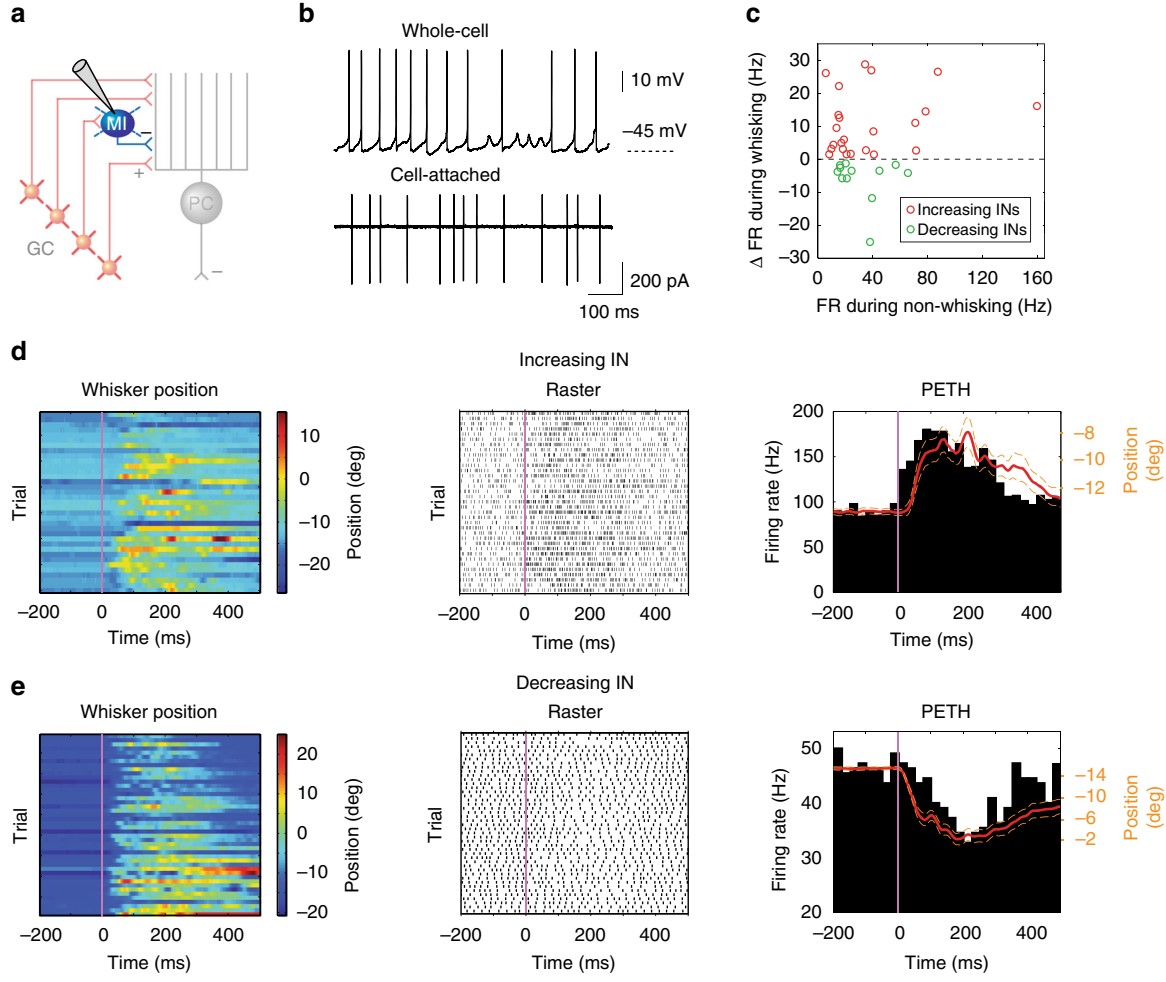

**Fig. 6** Firing rate alteration of inhibitory interneurons during whisking. **a** Schematic representation of cerebellar circuit highlighting inhibitory interneurons (INs) in the molecular layer. **b** Representative whole-cell and cell-attached recordings from INs. **c** Relative firing rate change during whisking with respect to baseline rate during non-whisking for all significantly modulated INs ($P < 0.05$, Wilcoxon signed-rank test, $n = 35$). *Red* and *green* circles represent increasing ($n = 23$) and decreasing ($n = 12$) cells, respectively. **d** IN exhibiting increasing firing rates during whisking. *Colour-coded* whisker movement (*left*) and the corresponding spike raster plot (*middle*). PETH from the same IN was overlaid with the mean whisker position (*red*; *dashed orange lines*: S.E.M) to show the close relationship between firing rate change and movement (*right*). **e** IN exhibiting decreasing firing rates during whisking. *Colour-coded* whisker movement (*left*) and the corresponding spike raster plot (*middle*). PETH from the same IN was overlaid with the mean whisker position (*red*; *dashed orange lines*: S.E.M) to show the close relationship between firing rate change and movement (*right*)

whisker[8]. Integration of afferent inputs from functionally distinct origins, e.g., ex-afferent sensory input during touch, in addition to efferent motor copy relating to other aspects of movement, such as locomotion, might be required to promote firing in the approximately two-thirds of Crus 1 GCs that are not active during whisker movement alone.

INs exert strong modulation over other cell types (and each other), placing them in a critical position to control information flow within cerebellar circuits. By making direct electro-physiological recordings from single INs in behaving mice, our experiments demonstrate INs in lobule Crus 1 show linear changes[40] in firing rate during voluntary whisking. Given that MLIs outnumber PCs by a factor of ten[52] while GoCs are relatively sparse, we consider that the majority of our INs recordings are MLIs. However, we were unable to identify sub-classes on the basis of electrophysiological characteristics (Supplementary Fig. 4), so our IN population may include Golgi, basket and stellate cells.

GC populations represent normalised set point information via monotonic increases in firing rate, i.e., movement either in front or behind the resting position is associated with significant

increases in spiking activity. In contrast, some INs, along with PCs[17, 53] solely exhibit reduced firing rates (and bidirectional PCs and INs exhibit both increases and decreases; Fig. 7b). These firing rate reductions cannot be explained by a pure excitatory drive from GCs, and is likely to result from local inhibition mediated through the action of MLIs[24, 38, 54] and/or PCs via collaterals[55]. When driven by elevated excitatory input from GCs, MLIs can in turn generate inverted responses in PCs and other postsynaptic INs via rapid feedforward and lateral inhibition[14–16, 23, 24, 26, 54, 56–59]. Molecular layer inhibition has been shown to implement sign reversal upon neighbouring adjacent PCs[24, 41], and our results suggest that this mechanism is a prominent feature of cerebellar encoding of whisking in awake, behaving animals. Our results indicate that, rather than receiving bidirectional changes in whisker-related MF activity, the cerebellar cortex splits behavioural signals via its inhibitory circuits, giving rise to bidirectional firing rate changes in output neurons[17, 53]. The di-synaptic GC-MLI-PC pathway thus provides a neural substrate to counterbalance monotonically increasing excitation, and allows MF inputs to undergo sign inversion (i.e., increases and decreases in firing rate between

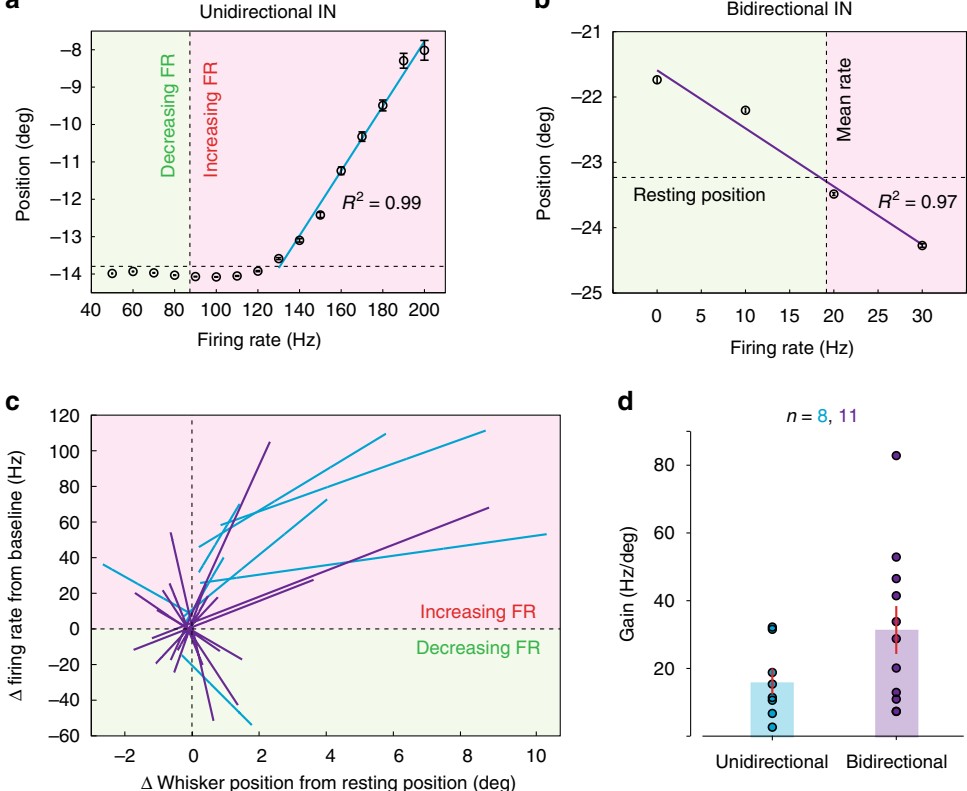

**Fig. 7** Linear relationships between inhibitory interneuron firing rates and whisker position. **a**. Example of unidirectional IN demonstrating piece-wise linear relationship between instantaneous firing rate and mean whisker position. Linear regression (in *blue*, $R^2 = 0.99$, Degrees of freedom = 7, F-test = 526.1, $P < 0.001$, ANOVA) was performed over the range of firing rate modulation by whisking (protraction in this case). *Vertical dashed line* indicates the cell's baseline rate during non-whisking. *Red* and *green* shading represent increase and decrease in firing rate, respectively. *Horizontal dashed line* shows the resting position of whisker. **b** Example of bidirectional IN. Linear fit (in *purple*, $R^2 = 0.97$, Degrees of freedom = 3, F-test = 70.8, $P < 0.05$, ANOVA) encompassing both directions of whisker movement. This cell decreased firing rate when the whisker was in the protraction side and increased rate while whisker moved in the retraction side. **c**. Summary of all unidirectional (*blue*, n = 8) and bidirectional (*purple*, n = 11) INs that demonstrated significant linear correlations between firing rate and whisker position ($R^2 = 0.96 \pm 0.01$, n = 19; $P < 0.05$, ANOVA). Relative changes in firing rate and whisker position were normalised with respect to each cell's spontaneous rate during non-whisking and the resting position of whisker, respectively. **d** Gain (Hz/deg), defined as the absolute value of the slope of individual linear regressions in **c**. for two types of INs ($P = 0.12$, Mann–Whitney U test)

populations of neurons) in downstream PCs[24]. In addition to reciprocal connectivity between interneurons, PCs collaterals also project back to neighbouring PCs and MLIs in the adult cerebellum[55]. This feedback projection could likewise offer an additional route for sign inversion. Together, GC-IN-PC, GC-IN-IN, GC-PC-PC and GC-PC-IN pathways may act in concert with the GC-PC connection to determine the firing rate change of PCs and INs in a bidirectional manner.

In contrast to GCs, INs exhibit quite distinct tuning properties with respect to whisker position. The most prevalent functional IN class consists of 'unidirectional' cells that exhibit altered firing rates at only forward or backward positions. This property is surprising, given that, across the entire population, upstream GCs encode both forward and backward positions via elevated firing. It is therefore possible that unidirectional INs receive excitatory inputs from specific subsets of GCs (i.e., forward- or backward-selective GCs), while 'bidirectional' INs may also receive inhibitory input from local unidirectional INs. The circuit mechanisms underlying this organisation remain unclear, and functional imaging of IN populations may reveal whether such functional heterogeneity is spatially organised (e.g., 'forward-' and 'backward-' movement microzones within Crus 1). At present, our results highlight the surprising complexity of function exhibited by cerebellar interneurons.

The role of the cerebellar cortex during sensorimotor behaviour has been widely debated. During voluntary whisking, a majority of PCs cells encode movement via linear changes in firing rate to represent salient kinematic parameters[17, 53, 60]. However, it has remained unclear how such movement-related signals propagate through the circuitry of the cerebellar cortex. The present study reveals that linear encoding is prominent in INs upstream of PCs, and indicates that broad sampling of PF inputs underpins the linear modulation of firing rate in INs and PCs that encode set point.

Our results provide a platform from which to address the crucial role of the lateral cerebellum in active sensory processing. Compared to multi-joint limb movements, whisking is a relatively simple behaviour, and we have focused on the movement with a single degree of freedom (forward and backward), though it is known that animals can also move single whiskers in three dimensions[61].

Beyond the representation of whisker position, cerebellar circuits also encode other kinematic features of whisking, including phase, which is robustly represented by neurons in somatosensory neocortex. Given the prominent sensory input received by Crus 1 (from both neocortex and periphery), a necessary next step is to determine the influence of external sensory input (e.g., tactile stimulation) on the activity of

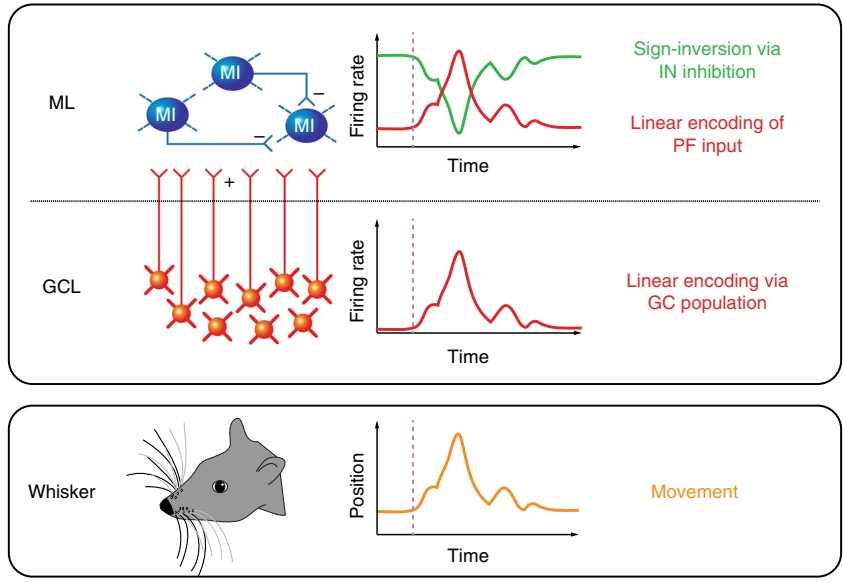

**Fig. 8** Serial processing of whisking signals within the cerebellar cortex. Voluntary movement (*orange* trace; *lower box*) is encoded by GC populations via linear increases in spike output (*red* trace; *upper box*) to represent whisker position. Excitation from GCs (via PFs) drives elevated firing rates in the molecular layer during whisker movement. Local inhibitory networks mediate sign-reversed responses (decreasing activity during whisking) in neighbouring INs (*green* trace; *upper box*). *GCL* granule cell layer, *ML* molecular layer, *MI* molecular layer interneuron, *PF* parallel fibre

cerebellar neurons during active whisking. Furthermore, it will be essential to record from different components of the cerebellar cortex during the acquisition and performance of vibrissal-based sensorimotor tasks[62–65], to fully establish the importance of the cerebellum in sensorimotor learning.

## Methods

**Animal handling and surgery**. The care and experimental manipulation of animals was performed in accordance with institutional and UK Home Office guidelines. 4- to 8-week-old C57BL/6 mice of both genders were used in this study. Animals were group housed in a 12–12 reverse light-dark cycle and all experiments were carried out during the dark phase. Prior to electrophysiology experiments, mice were anesthetised with 1–2% isoflurane under aseptic conditions, and a lightweight head-post was attached to the skull using glue Histoacryl (Braun Corporation, USA) and acrylic dental cement (Kemdent, UK). A circular chamber was built with cement over the lateral hemisphere of the cerebellum to allow subsequent access for electrophysiological recording. The chamber was then filled with a silicone-based elastomer (Kwik-Cast; World Precision Instruments, USA) and sealed with a layer of nail varnish. A non-steroidal anti-inflammatory drug (Carprofen; 5 mg/kg) was provided via intra-peritoneal administration during surgery to support recovery. Implanted mice were given 2–5 days for recovery, during which time Buprenorphine (0.8 mg/kg) jelly was provided for postoperative analgesia. On the day of the recording, anaesthesia was induced and a small cranial window (1–1.5 mm) was drilled over lobule Crus 1. The dura was removed with fine forceps and the craniotomy was covered with 1.5% low-melting point agar dissolved in cortex buffer (150 NaCl, 2.5 KCl, 10 HEPES, 2 CaCl₂, 1 MgCl₂ in mM) and subsequently sealant (Kwik-Cast; WPI, USA). Ipsilateral whiskers were partially trimmed with one whisker row left untouched (row C or D). At least 2 h following these procedures, habituation and recording sessions were started. Mice were carefully placed on a cylindrical treadmill and the head-post was gently loaded into fixation clamp for painless immobilisation of the head. At least 1 h of habituation was allowed for the mice to be acclimated to the recording environment. Habituated mice showed normal grooming, whisking and locomotion behaviours on the treadmill. After removal of sealant and agar, recordings were performed in the dark in a single session lasting up to 3 h[17]. Randomisation and blinding were not appropriate, given the nature of the experiments, and were not performed.

**In vivo electrophysiology**. Patch clamp recordings were made from cerebellar GCs and interneurons in awake mice using a Multiclamp 700B amplifier (Molecular Devices, USA). Recordings were made between depths of 200–1000 μm from the pia surface using borosilicate glass pipettes (6–8 MΩ) filled with internal solution containing (in mM): 135 K-gluconate, 7 KCl, 10 HEPES, 10 phospho-creatine, 2 Mg-ATP, 2 Na2-ATP, and 0.5 Na₂-GTP (pH 7.2, 280–290 mOsm). The data were filtered at 10 kHz, digitised at 25 kHz using an ITC-18 interface (Instrutech Corporation, USA) and transferred to a computer using AxographX

software (www.axograph.com). In whole-cell recordings, resting membrane potentials were recorded immediately after formation of whole-cell configuration and series resistances ranged between 20–40 MΩ. Bridge balance in current-clamp mode was applied in interneurons, but not in granule cell recordings, given their much higher input resistances in comparison to series resistances[25, 66]. No current was injected unless otherwise stated. Membrane potentials were not corrected for liquid junction potentials. Neuron types in the cerebellum (GCs and INs) could be readily identified by their characteristic electrophysiological properties[25, 35, 41].

**Whisker tracking**. Under infrared light illumination, whisker movements were filmed with a high-speed camera (Genie HM640; Teledyne Dalsa Inc, USA) operating at 250 fps Video acquisitions were controlled by Streampix 6 software (Norpix, Canada) and externally triggered by TTL pulses generated via the ITC-18 to synchronise with electrophysiological acquisition. Whisker position was tracked offline using open-source software[67]—http://whiskertracking.janelia.org—and a customised graphical user interface in MATLAB (Mathworks). Whisker azimuth angles were measured along the longitudinal axis (medial line: 0 degree); protraction corresponded to increases in whisker angle. Because whiskers, especially those from the same row, move in synchrony, one of the traced whiskers was routinely used for analysis concerning whisking, as changing whisker did not affect the results[68].

**Data presentation and analysis**. The data are presented as mean ± SEM unless otherwise stated. All the data analysis was carried out in MATLAB (MathWorks) and Axograph. In whole-cell recordings, input resistance of GCs was measured from steady-state voltage deflections during 400 ms step current injections of –20 pA. Interneuron input resistance was calculated using 400 ms current injection of –50 pA. GC synaptic events were detected using a template-matching algorithm in Axograph X, where a representative EPSC event was selected to serve as the template in each cell and event detection was set at least three times the standard deviation of the baseline noise. All detected EPSCs were visually inspected. Cell-attached recordings were first high-pass filtered at 20 Hz. Action potentials were detected automatically using an amplitude threshold in Axograph X. Sample size estimation was not performed due to the technically challenging nature of the recordings and instead post hoc tests were performed to assess statistical significance.

**Behavioural characterisation**. Whisking epochs were visually identified off-line. Traced whisker position was first low-pass filtered at 30 Hz using a 4-pole Butterworth filter run in forward and reverse directions, and subsequently up-sampled to 1 kHz. Kinematic feature set point was derived by low-pass filtering whisker angle at cutoff frequency 6 Hz[69]. Kinematic feature phase was defined as the angle of the Hilbert transform on band-pass filtered (6–30 Hz) whisker angle. A phase of zero corresponds to maximal protraction and a phase of ±π denotes maximal retraction in a whisk cycle[68].

**Granule cell analysis**. Spikes were grouped into bursts with inter-spike interval (ISI) > 50 ms. Wilcoxon signed-rank test ($P < 0.001$) was used to evaluate significant change

in firing rate or EPSC rate across the population of GCs. Significant changes in membrane potential distribution between non-whisking and whisking epochs were determined using a 2-sample Kolmogorov–Smirnov test ($P < 0.05$). Membrane potential and the whisker position data were truncated into 1 s segments centred on individual whisking onsets (0.5 s preceding- and post-onset). To examine the temporal relationship between membrane potential depolarisation and whisker position change, normalised cross-correlations were computed for the individual data segments and averaged across segments. The time at the nearest maxima (peaks) above the upper/lower 95% confidence bounds defined the time delay between the two signals.

To identify GC tuning to kinematic features, a two-sample Kolmogorov–Smirnov test ($P < 0.05$) was used to compare the distribution of angle (position/set point) at all times with the distribution at times of EPSC/spike occurrence. The distribution of kinematic features at EPSC/spike times was normalised by the amount of time spent in individual bins to generate the tuning curve (in terms of Hz). Kuiper's test ($P < 0.05$) was used to assess phase tuning. Modulation depths were calculated as the maximal rate (preferred) minus the minimal rate (least-preferred) divided by the mean in the tuning curve[68]. The firing rate tuning curves of individual significantly modulated GCs were normalised into percentiles in 5% increments with respect to the range of set point in individual mice. A population set point tuning curve was generated via averaging across cells for individual percentiles.

**Interneuron analysis**. Spike rates were calculated across all whisking and non-whisking epochs in the recording sweeps as the total number of spikes divided by the duration of an epoch. CV2 of ISI was calculated using $CV2_n = 2 \times (ISI_{n+1} - ISI_n)/(ISI_{n+1} + ISI_n)$. Comparisons of the spike rates were made between quiet and whisking epochs using the non-parametric Wilcoxon signed-rank test, where $P < 0.05$ was recognised as a significant difference[70]. Overall firing rates during whisking and non-whisking were calculated by averaging the spike rates of all epochs comprising the two respective conditions. To generate peri-event time-histograms, spike trains were aligned by the onsets of whisking bouts and averaged across trials. Corresponding whisking epochs were aligned at the onset and averaged to reveal the mean whisker movement within bouts.

To determine instantaneous firing rates, a 100 ms rectangular window function was slid along IN spike train with 1 ms step. All linear regression fits for IN instantaneous firing rate and average whisker position was performed using the Basic Fitting GUI in MATLAB. Analysis of Variance (ANOVA) was used to determine whether variation in whisker position arises among different instantaneous firing rate groups of a given IN, and a $P$-value smaller than 0.05 justified linear modulation of firing rate by whisker position.

**Linear decoding by transfer function**. The relationship between the IN firing pattern and whisker position was modelled by a linear transfer function, which was calculated in order to decode whisking trajectory from single neuron spike trains. Open-source Chronux Software (http://chronux.org/) applied in MATLAB was used to generate transfer functions by multi-taper estimation described in detail elsewhere[68]. Briefly, using 10–20 s of training data set with both quiet and free whisking epochs, the transfer function $H(f)$ was calculated from the Fourier transform of the spike train $S(f)$ as a sequence of Dirac delta functions, and the Fourier transform of the whisker position $\theta(f)$ via

$$H(f) = \frac{\langle S(f)\theta(f)^* \rangle}{\langle S(f)^2 \rangle}$$

in which the asterisk denotes complex conjugate and the angular brackets indicate average across tapers and trials. The computed transfer function was then applied to the Fourier transform of the raw spike train in test trial $S(f)_{test}$ to reconstruct the Fourier transform of the whisker trajectory $\theta_{reconstruct}(f)$ by

$$\theta_{reconstruct}(f) = H(f)S(f)_{test}$$

The inverse Fourier transform of this function is the predicted whisker motion. Pearson's correlation coefficients were calculated between the reconstructed whisker position and the real whisker set point derived from the original whisker trace.

**Data availability**. All the relevant data are available from the authors.

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

## Acknowledgements

We thank Claudia Clopath for support and feedback on the manuscript. This work was supported by a NUS Graduate School scholarship (SC), a CRP grant from the National Research Foundation of Singapore (GJA), a UK Medical Research Council Career Development Award (G1000512) and grants from the Human Frontier Science Program and the Biotechnology and Biological Science Research Council (PC).

## Author contributions

S.C., G.J.A. and P.C. designed the experiments. All experiments were performed by S.C. The data were analysed and interpreted by S.C. and P.C. S.C. and P.C. wrote the manuscript with contributions from all authors.

## Additional information

**Competing interests:** The authors declare no competing financial interests.

