## [Peer Review File · Nature Communications]

Reviewers' Comments:

Reviewer #1 (Remarks to the Author):

In this study, Chen and colleagues performed a remarkable series of recordings of granule cells (one of the main processing stages of the cerebellum but also one of the most elusive type of neurons in the brain) and downstream interneurons in behaving animals. They were able to correlate both the inputs (EPSP) and output (spiking activity) of granule cells with whisking activity, and also described IN firing during whisking.

The results are clear and well presented and the statistical analyses are appropriate and convincing.

This important study provides an original picture of how GC behave in vivo and will be of broad interest for the scientific community.

Major remarks:

Paragraph: 'Reconstruction of whisker set point but not phase from single IN activity': there are a couple of issues with this section. First, in the example (Fig 6e), the predicted whisker position has a much lower amplitude than the actual whisker position, as indicated by the difference in the scale bar (2 deg for the predicted whisker position versus 10 deg for the actual position). However, the dynamic is correctly predicted. This suggests that the authors trained their decoder with a set of data where the neuron's gain was not the same than in the segment of data presented here. Altogether, this example may confuse the reader.

Furthermore, the author have already shown that the firing rate of many IN is linearly related to whisker position. Therefore, it is clear that the firing of IN should be able to predict whisker position and presenting decoding results seems to be of secondary interest. If the authors wish to keep this decoding part, I suggest that they find a better example IN or fix this gain issue.

Minor remarks:

Abstract: please mention the recording location (Crus 1) in the abstract.

Legend of Fig 2c is very unclear. What are different movement epochs? Are a 'movement epoch' and a trial the same thing?

It seems that you are showing a single GC in Fig 2c, although the corresponding text ('Therefore despite only receiving a handful of MF inputs, individual GCs precisely represent whisker position during epochs of voluntary movement via subthreshold fluctuations in membrane potential') sounds like you refer to the entire GC population. If Fig. 2c indeed shows a single GC, then it should be clear in the text.

How do the authors know that they recorded from Molecular Layer and not Granular Layer interneurons? They argue (lines 352-354) that MLI outnumber GoCs. However, one could object that GoCs are easier to isolate than MLI. I suggest that the authors discuss the identification of their IN in more details.

Could the authors provide a scatterplot that represents the CV2 versus average firing rate (during resting) of the neurons they recorded?

Reference 39 needs to be fixed (the authors first and last names are mixed).

Reviewer #2 (Remarks to the Author):

This MS follows on from a previous article in eLife by the same authors investigating the relation between the activity of cerebellar Purkinje cell firing and whisking. There, the authors described a remarkable linear encoding of whisking set-point. In this paper, the authors perform very similar experiments, but examine the activity of the two downstream neuronal types: granule cells and molecular layer interneurons. Granule cells display a rather noisy, nonlinear positional code - high activity is recorded at a particular whisker position, while average activity seems to be approximately piece-wise linear. In contrast, interneurons individually show strongly linear position codes that recall those of Purkinje cells.

I find the work to be of high quality. It is also quite interesting, although perhaps not quite so much as the previous paper it seeks to explain. A few aspects of the analysis and presentation were not clear to me, as detailed below.

Presumably the higher spontaneous rate in cell-attached recordings can be explained by the fact that there was a selection bias against silent granule cells? Is this explained somewhere? The cell-attached recordings aren't detailed in the methods.

The number of whole cell recordings from responsive granule cells is low (n=5) or have I misinterpreted the statement that 10/15 granule cells were silent throughout the recording?

Line 140: how many cells?

Line 165: I don't really agree that individual granule cells precisely represent whisker position.

Is it possible to show that the somewhat undramatic changes of EPSP peak, frequency and rise time can account for the observed firing changes? Might other mechanisms be involved, such as altered inhibition?

In the reconstruction of position from individual INs, how do the authors deal with the firing range where unidirectional cells couldn't care less about whisker position?

What is remarkable about the unidirectional INs is that whatever causes their firing to vary in the range in which they are insensitive to position is completely absent in the range where they are. What is the authors explanation for this?

Line 348: a bit confusing to say that INs exert strong modulation over GCs (do you mean Golgi cells?).

The analysis description line 525+ is hard to understand. I'm lost at the KS test - what is it for?

I haven't yet understood the dashed lines Fig. 4A. Maybe the authors could help me and other readers.

Reviewer #3 (Remarks to the Author):

General Comments

This manuscript is an interesting study on the modulation of cerebellar granular cells and inhibitory interneurons during whisking in the mouse. The recordings were done with either whole-patch or cell-attached, and therefore the approach is both challenging and powerful. The patch clamp recordings have the potential to inform us about sub-threshold events in the cerebellar circuit. The main findings are bidirectional increases in granule cell discharge during whisking while the interneuron responses are more complex. Another very interesting observation is that the depolarization in granule cells leads whisker movement. While the results are interesting, there are several main concerns. Until these concerns are addressed, the impact of the work is not clear.

First, the results present a less than clear picture of how whisking is encoding. Because the granule cells respond with increased discharge in both directions of whisking, it is difficult to understand the nature of the signals encoded. It appears that only amplitude of whisker movement but not direction is encoded. This seems like a very poor way to provide the types of information that is needed. The question is how is this information is disambiguated in the rest of the circuit? To this reviewer, the recordings and analysis of the cerebellar interneurons does not greatly clarify this critical question.

Second, the presentation and statistical analyses of the results is unsatisfactory and needs considerable clarification if the reader is to comprehend and evaluate the results. Four statistical

problems are common throughout the manuscript and are highlighted here: 1) Many of the plots of various results do not provide any measure of response variability. Some examples include the plots in Figure 1C, D and E. Each plot of firing or the EPSP rate etc. of an individual cell should not just have a mean but also standing deviation plotted. Similar concerns are for the mean whisking position plots in Figure 2a. These are just examples of the problem, and more can be found throughout the manuscript. 2) The analyses makes use of linear regression. However, most of these regressions appear to be done on the mean values and not the individual data points (if not, this needed to be clarified). Regressions on the mean values inflate the R^2 values and should be avoided. Also, for the R^2 values provided there is no F-test, degrees of freedom, or p-values provided on the significance of the coefficient of determination. This information needs to be included. 3) Many of the results are presented as single examples (e.g. Figure 3d) and population results should be provided for all key observations. 4) The “n” for the granule cell recordings vary from 28, 26, 8, and 9 in different analyses. The number of cells is confusing and needs clarification. 5) The decoding analysis presented at the end of the results is interesting but needs to be fleshed out much more. The correlation provided for the strength of the decoding is not accompanied by any measure of significance.

A final general comment is that there is no mention of inhibitory synaptic inputs for either the granule cell or inhibitory interneuron recordings. Does this imply that Golgi cell inhibition is not operative? Do the recordings show the inhibitory input from Purkinje cells or other interneurons is operative on the molecular layer of inhibitory neurons? As noted above, the power of the approach is to record sub-threshold events and developing this aspect of the findings would strengthen the manuscript.

Specific Comments

- 1) The results presented in Supplementary Figure 1 are not particularly convincing. For example, only 3 cells in 1b actually increase their EPSP rate with whisking. The cumulative probability differences in 1c are small, is only shown for 4 cells and not particularly compelling. The plots in 1d are only for 2 cells, again not convincing.
- 2) A key argument and potentially one of the most important findings is that granule cell depolarization occurs before the onset of whisking. However, granule cell firing does not appear to occur prior to whisker movement. This dissociation needs to be addressed. The early depolarization is potentially the most important observation, but needs to be fully developed and supported.
- 3) Another issue is that several of the whisking position plots, Figure 5d and e, show a delay in the whisking position. Can this delay account for the early depolarizations in the granule cells?
- 4) The set-point analysis is not well described and should be clarified.

Responses to Reviewers – Chen *et al.*

We thank the Reviewers for their feedback and comments regarding the quality and interest of our work. We have performed additional experiments and analysis to address the questions and criticisms that have been raised and have greatly enhanced the impact of the study.

We have performed a new set of voltage-clamp recordings from granule cells in awake behaving mice ($n = 11$ cells) to measure excitatory postsynaptic currents and demonstrate how MF synaptic input is modulated by movement. We have also performed additional voltage recordings (i.e. current-clamp; $n = 17$) to clarify the relationship between GC membrane potential and spike output.

Throughout the manuscript, we have revised the presentation and statistical analysis of our results. We have added a new data figure (Figure 2), and substantially revised Figures 1, 3 and 4. Together, we provide a clear picture of how movement signals arriving at the input layer of the cerebellum are processed by the smallest cells in the brain. We go on to show that sign inversion via inhibitory interneurons is essential to explain the linear increases and decreases of Purkinje cell spike rate during whisking.

Reviewers' comments:

Reviewer #1 (Remarks to the Author):

In this study, Chen and colleagues performed a remarkable series of recordings of granule cells (one of the main processing stages of the cerebellum but also one of the most elusive type of neurons in the brain) and downstream interneurons in behaving animals. They were able to correlate both the inputs (EPSP) and output (spiking activity) of granule cells with whisking activity, and also described IN firing during whisking.

The results are clear and well presented and the statistical analyses are appropriate and convincing.

This important study provides an original picture of how GC behave in vivo and will be of broad interest for the scientific community.

Major remarks:

Paragraph: 'Reconstruction of whisker set point but not phase from single IN activity': there are a couple of issues with this section. First, in the example (Fig 6e), the predicted whisker position has a much lower amplitude than the actual whisker position, as indicated by the difference in the scale bar (2 deg for the predicted whisker position versus 10 deg for the actual position). However, the dynamic is correctly predicted. This suggests that the authors trained their decoder with a set of data where the neuron's gain was not the same than in the segment of data presented here. Altogether, this example may confuse the reader. Furthermore, the author have already shown that the firing rate of many IN is linearly related to whisker position. Therefore, it is clear that the firing of IN should be able to predict whisker position and presenting decoding results seems to be of secondary interest. If the

authors wish to keep this decoding part, I suggest that they find a better example IN or fix this gain issue.

We have looked further into this issue and found that this is actually not a gain problem, as movement amplitudes are always reduced in the reconstruction. It seems the reconstruction provides an attenuated ‘low-pass’ version of the actual movement as the amplitude is reduced and the temporal dynamics are slowed in the reconstruction. This result is partly due to the fact that we include the full behavioural range of whisking (i.e. including positions at which individual neurons are unresponsive or saturated). It may also be related to the fact that individual INs have lower firing rates than Purkinje cells, limiting their ability to represent fast changes via firing rate alone. INs also receive a smaller number of synaptic inputs than Purkinje cells, so this may reduce the resolution of individual INs.

We have addressed this issue in the manuscript in two ways: first, we have moved the reconstruction to Supplementary Figure 5. Second, we specifically highlight the attenuated nature of the reconstructed signal in this figure, and in the Results (line 308-312) and conclude the INs provide only a partial representation of whisker set point.

Minor remarks:

Abstract: please mention the recording location (Crus 1) in the abstract.

Added at line 32.

Legend of Fig 2c is very unclear. What are different movement epochs? Are a ‘movement epoch’ and a trial the same thing?

Trials are individual movement epochs. We now state this in the legends of Figures 2 & 4.

It seems that you are showing a single GC in Fig 2c, although the corresponding text (‘Therefore despite only receiving a handful of MF inputs, individual GCs precisely represent whisker position during epochs of voluntary movement via subthreshold fluctuations in membrane potential’) sounds like you refer to the entire GC population. If Fig. 2c indeed shows a single GC, then it should be clear in the text.

We performed additional experiments to better characterise how individual GCs encode whisker movement. We plot the relationship between kinematic parameters and membrane potential in detail for two representative cells (Figure 3b-c and Figure 4e-f), and quantify the range of movement-related V_m modulation for the entire population ($n = 26$; Figure 3d). We believe these data are now appropriate to support our original statement.

How do the authors know that they recorded from Molecular Layer and not Granular Layer interneurons? They argue (lines 352-354) that MLI outnumber GoCs. However, one could object that GoCs are easier to isolate than MLI. I suggest that the authors discuss the identification of their IN in more details.

We are unable to rule out that a minority of our interneuron recordings are from the granular layer. However, the majority of these recordings were targeted to superficial depths in the cortex, and their electrophysiological properties closely match recordings from MLIs (e.g. Jelitai *et al.*, 2016). We have added additional information about IN classification in the Results section (the scatterplot of CV2 vs. firing rate in a new Supplementary Figure 4 as requested below), and discuss the caveats of our approach (line 408-412).

Could the authors provide a scatterplot that represents the CV2 versus average firing rate (during resting) of the neurons they recorded?

We have added this information in Supplementary Figure 4.

Reference 39 needs to be fixed (the authors first and last names are mixed).

Done.

Reviewer #2 (Remarks to the Author):

This MS follows on from a previous article in eLife by the same authors investigating the relation between the activity of cerebellar Purkinje cell firing and whisking. There, the authors described a remarkable linear encoding of whisking set-point. In this paper, the authors perform very similar experiments, but examine the activity of the two downstream neuronal types: granule cells and molecular layer interneurons. Granule cells display a rather noisy, nonlinear positional code - high activity is recorded at a particular whisker position, while average activity seems to be approximately piece-wise linear. In contrast, interneurons individually show strongly linear position codes that recall those of Purkinje cells.

I find the work to be of high quality. It is also quite interesting, although perhaps not quite so much as the previous paper it seeks to explain. A few aspects of the analysis and presentation were not clear to me, as detailed below.

Presumably the higher spontaneous rate in cell-attached recordings can be explained by the fact that there was a selection bias against silent granule cells? Is this explained somewhere? The cell-attached recordings aren't detailed in the methods.

Yes, selection bias against silent cells accounts for the difference in spontaneous firing rate. A statement clarifying this point has been added to the Results (line 127-131). If we remove silent cells and compare CA and WC firing rates, they are not significantly different.

The number of whole cell recordings from responsive granule cells is low (n=5) or have I misinterpreted the statement that 10/15 granule cells were silent throughout the recording?

Our revised count is 32 GCs recorded in current-clamp (voltage recording) mode, of which 20 were silent. This is stated in line 131.

Line 140: how many cells?

These values were meant to read $n = 13$ cell-attached, and $n = 15$ whole-cell in the previous version. With additional current clamp recordings, the whole-cell count is now 32 and this is stated on line 113.

Line 165: I don't really agree that individual granule cells precisely represent whisker position.

We have performed additional analysis to provide more evidence for this claim, however we accept that GCs only precisely represent whisker position within a very limited range. We have removed this statement and revised the text of the Results accordingly.

Is it possible to show that the somewhat undramatic changes of EPSP peak, frequency and rise time can account for the observed firing changes? Might other mechanisms be involved, such as altered inhibition?

We have performed additional experiments to deal with these questions. To get around issues such as the influence of driving force on EPSP amplitude, we have made voltage clamp recordings at -70mV to resolve EPSCs. This dataset demonstrates that EPSC amplitude and time-course are essentially unchanged during whisking whereas EPSC rate is strongly altered. This information forms the basis of a new Figure 2. In a subset of neurons, we were able to do voltage clamp and V_m recording consecutively. In these cells, we show that the kinematic tuning of EPSC rate is identical to the tuning of membrane potential. This result allows us to rule out a role for synaptic inhibition in shaping the subthreshold tuning of individual GCs, and is presented in a new Figure 3. Finally, we show that output tuning corresponds to the strongest region of the subthreshold tuning curve (presented in new Figure 4). Overall these new data show that the rate of excitatory MF synaptic input is by far the most important factor determining which features of movement individual cells will respond to. We have substantially revised the text of the results and discussion to include this information, and also discuss what the potential role(s) of GCL inhibition may be in this context.

We believe this information has substantially improved the manuscript and greatly strengthened our conclusions.

In the reconstruction of position from individual INs, how do the authors deal with the firing range where unidirectional cells couldn't care less about whisker position?

To generate the transfer function, we take 10 – 20 second epochs of movement data and the corresponding IN spike train. We then use this function to reconstruct another tranche of movement using the spike train alone. Qualitatively we see that the reconstruction only recovers trajectories within the linear range for each neuron. It would be possible to limit the whisking range over which to attempt reconstruction and improve reconstruction quality (i.e. restrict only to positions at which each neuron responds). However, this would need to be done on a cell-by-cell basis and we have chosen instead to quantify the reconstruction

quality to *all* of the behaviour, and provide an overall picture of how well individual neurons encode behaviour.

What is remarkable about the unidirectional INs is that whatever causes their firing to vary in the range in which they are insensitive to position is completely absent in the range where they are. What is the authors explanation for this?

We agree that this is indeed remarkable. We do not have a definitive explanation, we currently think this result (as for Purkinje cells) suggests that there may be a modular organisation of movement signals in the ML/PC (and potentially also the underlying granule cell layer). As the reviewer points out, a forward-selective ‘unidirectional’ cell does not change its firing rate at all to backward movement. This may mean that these cells do not receive any notable synaptic input from GCs that encode backward movement, or alternatively such input is cancelled via inhibition. We now highlight this issue in a new section of the Discussion (‘Complex positional tuning of inhibitory interneurons’; line 437).

Line 348: a bit confusing to say that INs exert strong modulation over GCs (do you mean Golgi cells?).

We have reworded this sentence to minimise confusion (line 404).

The analysis description line 525+ is hard to understand. I'm lost at the KS test - what is it for?

The aim of this analysis was to detect whether aspects of GC activity such as EPSC- or action potential rate were significantly modulated with respect to kinematic features of whisking. We compared the distribution of one kinematic feature (e.g. set point) at all times to its distribution at times of EPSC/spike occurrence during whisking. For position and set point, we tested whether the two distributions were different using a 2-sample Kolmogorov-Smirnov test. For circular variable phase, we used a 2-sample Kuiper’s test. This information has been revised in the Methods (line 567-572).

I haven't yet understood the dashed lines Fig. 4A. Maybe the authors could help me and other readers.

The dashed lines indicate the whisker resting position. This information has been added to the legend of the corresponding figure in this version (Figure 5a).

Reviewer #3 (Remarks to the Author):

General Comments

This manuscript is an interesting study on the modulation of cerebellar granular cells and inhibitory interneurons during whisking in the mouse. The recordings were done with either whole-patch or cell-attached, and therefore the approach is both challenging and powerful. The patch clamp recordings have the potential to inform us about sub-threshold events in the cerebellar circuit. The main findings are bidirectional increases in granule cell discharge

during whisking while the interneuron responses are more complex. Another very interesting observation is that the depolarization in granule cells leads whisker movement. While the results are interesting, there are several main concerns. Until these concerns are addressed, the impact of the work is not clear.

First, the results present a less than clear picture of how whisking is encoding. Because the granule cells respond with increased discharge in both directions of whisking, it is difficult to understand the nature of the signals encoded. It appears that only amplitude of whisker movement but not direction is encoded. This seems like a very poor way to provide the types of information that is needed. The question is how is this information is disambiguated in the rest of the circuit? To this reviewer, the recordings and analysis of the cerebellar interneurons does not greatly clarify this critical question.

We apologise for the lack of clarity in the previous manuscript. We have performed new experiments and analyses to tackle several aspects of the issue of how whisking is encoded. Individual granule cells are quite selective for position/set point (and also in a few cases, phase) – in other words they only spike, if at all, to a relatively small fraction of whisker angles and phases. We have quantified this in Figure 4 and Supplementary Figure 3. In particular, we show that granule cells are more sharply tuned in their output than input (Figure 4 e,f,h). We also revised Figure 5a to show 5 separate GC tuning curves and highlight the narrow fraction of whisker angles that each cell encodes. Given that individual downstream neurons are only likely to receive an input from a subset of GCs, this allows information to be disambiguated in the rest of the circuit. We now highlight this issue in a new section of the Discussion ('Complex positional tuning of inhibitory interneurons'; line 437).

Second, the presentation and statistical analyses of the results is unsatisfactory and needs considerable clarification if the reader is to comprehend and evaluate the results. Four statistical problems are common throughout the manuscript and are highlighted here:

1) Many of the plots of various results do not provide any measure of response variability. Some examples include the plots in Figure 1C, D and E. Each plot of firing or the EPSP rate etc. of an individual cell should not just have a mean but also standing deviation plotted. Similar concerns are for the mean whisking position plots in Figure 2a. These are just examples of the problem, and more can be found throughout the manuscript.

We have fully revised the Results section and individual figures to add measures of response variability wherever appropriate. We hope that this problem has now been resolved throughout the manuscript.

2) The analyses makes use of linear regression. However, most of these regressions appear to be done on the mean values and not the individual data points (if not, this needed to be clarified). Regressions on the mean values inflate the R^2 values and should be avoided.

Our regressions are on mean data, and we provide this information in the manuscript (line 296) and in a new Supplementary Table 1. We accept that regressions on mean data are

not ideal, but our measurements are made using relatively short periods of spontaneous behaviour in awake animals. We therefore have considerable sources of variability including variability of the whisker movement itself and the potential confound of neuronal activity driven by non-whisking movement (e.g. postural or facial movements). In order to extract meaningful relationships between whisker movement and single neuron activity, we have therefore performed averaging prior to measuring regressions. All information about individual linear regression models are included in Supplementary Table 1.

Also, for the R^2 values provided there is no F-test, degrees of freedom, or p-values provided on the significance of the coefficient of determination. This information needs to be included.

We have included this information for individual cell examples in the legend of Figure 6, and comprehensively in Supplementary Table 1.

3) Many of the results are presented as single examples (e.g. Figure 3d) and population results should be provided for all key observations.

We now provide population results for all key observations.

4) The “n” for the granule cell recordings vary from 28, 26, 8, and 9 in different analyses. The number of cells is confusing and needs clarification.

We have clarified this throughout the manuscript.

In summary:

GC whole-cell current clamp

32 recordings –

26 cells show a significant change in Vm during whisking; in 5 of these cells we also performed voltage clamp.

12 of these cells show action potential firing during whisking

GC voltage clamp

11 recordings; 5 of these cells are also included in current clamp group

5) The decoding analysis presented at the end of the results is interesting but needs to be fleshed out much more. The correlation provided for the strength of the decoding is not accompanied by any measure of significance.

We measured the correlation coefficient between the actual and reconstructed whisker trajectories. These relationships were all statistically significant ($p < 0.05$). This information has been added to the legend of Supplementary Figure 5.

A final general comment is that there is no mention of inhibitory synaptic inputs for either the granule cell or inhibitory interneuron recordings. Does this imply that Golgi cell inhibition is not operative? Do the recordings show the inhibitory input from Purkinje cells or other interneurons is operative on the molecular layer of inhibitory neurons? As noted above, the power of the approach is to record sub-threshold events and developing this aspect of the findings would strengthen the manuscript.

We accept that we have not been able to reveal the precise role of synaptic inhibition within cerebellar cortex. This is a very challenging task to achieve in awake animals as it is necessary to perform voltage clamp recording at both -70mV and 0 mV (to individually resolve EPSCs and IPSCs), and also record V_m in the same cell (to assess the combined influence of synaptic excitation and inhibition). Such experiments require very long recordings and have only recently been performed in anaesthetised animals (Duguid *et al.*, PNAS 2015).

Nevertheless, our voltage-clamp recordings do shed considerable light on the potential role(s) of Golgi cell inhibition during movement. We show clearly that the GC output tuning is tightly linked to the underlying kinematic tuning of EPSC rate. Thus we can conclude that Golgi cell inhibition does not play a prominent role of shaping the relative selectivities of individual GCs to kinematics. Of course this does not rule out other proposed roles for Golgi cells (e.g. rate and timing of GC output) and we discuss these issues, along with the role(s) of ML inhibition in new sections of the Discussion (from lines 363 and 437).

Specific Comments

1)The results presented in Supplementary Figure 1 are not particularly convincing. For example, only 3 cells in 1b actually increase their EPSP rate with whisking. The cumulative probability differences in 1c are small, is only shown for 4 cells and not particularly compelling. The plots in 1d are only for 2 cells, again not convincing.

We have replaced these plots with voltage clamp data from GCs. This has allowed us to use EPSC measurements that have far better temporal resolution, and are not subject to changes in driving force. We include this information in new Figures 2 & 3.

2)A key argument and potentially one of the most important findings is that granule cell depolarization occurs before the onset of whisking. However, granule cell firing does not appear to occur prior to whisker movement. This dissociation needs to be addressed. The early depolarization is potentially the most important observation, but needs to be fully developed and supported.

We measured the temporal relationship between whisking and GC activity using cross-correlation (Figure 1f) and find that depolarisation precedes movement in the majority of GCs (Figure 1g). This information is detailed from line 138 onwards. We do not have enough action potentials from individual recordings to accurately perform the same analysis for GC spiking. However, we include an example recording in Supplementary 1 that highlights granule cell spiking prior to movement onset.

3)Another issue is that several of the whisking position plots, Figure 5d and e , show a delay in the whisking position. Can this delay account for the early depolarizations in the granule cells?

Thank you for highlighting the apparent delay in these plots. We have inspected these traces and in both cases the onset of whisking is characterised by a very small retraction followed by a large amplitude protraction. The initial retraction was used to manually align whisking trials but is not really resolved in the colour plots. We have rechecked all of our data and cannot find observable delays in other whisking traces (including Fig 2b and 4b). The traces in Figures 5d and e (now 6d and e) were manually aligned for display purposes, but our latency calculations are performed using cross-correlation so are not sensitive to manual alignment of traces. Therefore, we are certain that this alignment does not account for early depolarisations in GCs.

4)The set-point analysis is not well described and should be clarified.

Set point is the slow positional change derived from low-pass filtering position signal at 6Hz. In order to establish whether GCs were significantly modulated by this parameter, we compared the distribution of set point at all times to its distribution at times of EPSC/spike occurrence during whisking. A similar analysis was previously performed by the Kleinfeld group (Hill *et al.*, Neuron 2011). We tested whether the two distributions were different using a 2-sample Kolmogorov-Smirnov test. This information has been revised in the Methods (line 567-572).

Reviewers' Comments:

Reviewer #1 (Remarks to the Author):

This is a revision of the manuscript "Serial processing of kinematic signals by cerebellar circuitry during voluntary whisking", where authors characterize the subthreshold activity and firing responses of cerebellar granule cells during whisking behavior. In this revision, the authors have addressed my previous comments and improved the clarity and impact of the manuscript. I have no further major remarks, and only minor comments.

This study offers a remarkable view of information flow through the cerebellum (i.e. at the input and output of granule cells, in the molecular layer and, in the author's previous study, in the Purkinje cells layer) during motor behavior. This work should be of high interest to a wide audience.

Minor comments:

Fig. 1c is difficult to read. The text states that 20 out of 32 cells are silent in WC mode. I assume that these cells collapse in a single circle at the bottom of the bar. Although 12 cells should have non-zero firing rate, I distinguish only 5 non-zero markers in the graph. Could the author improve this panel's presentation, or its description in the legend?

Fig. 4c,d: how do you define the firing rate? It appears that you compute PSTH with a bin size of 50ms but this is not indicated in the text or legend.

Reviewer #2 (Remarks to the Author):

The authors have performed additional experiments and otherwise made very careful efforts to respond to my comments.

I remain somewhat unimpressed by the population coding implied by the piecewise linear fit in Fig. 5b. However, the data are displayed for all to interpret as they wish, and I don't believe the precise coding of the granule cells will be fundamental, as the downstream cells can presumably select desired inputs via learning and plasticity. Maybe the authors could start the y-axis of this plot at zero?

Reviewer #3 (Remarks to the Author):

General Comments

Overall, the authors have made a concerted effort to address this reviewer's concerns as well as the concerns of the other two reviewers. The manuscript has improved and has many important

findings on cerebellar granule cells (GCs) and interneurons (INs) during whisking in the mouse. The fact that the study was using patch clamping adds greatly to the study. There are two general comments and several specific comments.

The first general comment is the value-added to examining the relation between the responses and whisker position and set point. Set point is simply a low pass filtered version of whisker position. The two measures are highly correlated as shown in Figure 3a. One suspects that if one did a cross-correlation between whisker position and set point, the correlation would be quite high. Therefore, the neural responses in relation to these two measures will be similar as shown in Figure 3c and the finding that the neurons are correlated with both position and set point is not surprising. The authors would have to demonstrate that the two measures are really providing independent information on the behavior or should consider removing/downplaying.

The second general comment is the argument that the modulation depth for the GC responses is smaller for the input (EPSC) than for the output (firing rate) as shown in Figure 4h. However, whether this comparison is valid needs to be proven. To be valid the comparison needs to take into account the range and variability of the two measures, normalizing in a way to minimize the differences. For example, the range and ceiling effect of the two measures are very different. EPSC amplitudes will have a small range and variability with a ceiling after which the cell will fire, while firing rate in GCs can go from 0 to +200Hz. Given these differences would it even be possible for the depth of modulation of the EPSCs to be greater than for firing rate?

Specific Comments

- 1) On line 129 should Fig. 1d be Fig. 1c?
- 2) In relation to Figure 4b, the text describes much higher bursting rates for GCs than the example selected. Why wasn't a more representative example of GC responses used? A similar question is for the spike rate in Figure 4c, that seems low when the burst rate is so high?
- 3) On page 9, the text states there 45 INs were recorded, Should the statement on decreasing firing stating $n=12/45$ instead of $n=12/35$?
- 4) Way was the reconstruction of the whisker trajectories done on the GC responses? It seems this would be very interesting comparison and help clarify the serial processing that is the theme of the study?

Final Revisions – Response to Reviewers (NCOMMS-16-21830B)

We again thank the reviewers for their comments, and have made edits to figures and legends on the basis of this feedback. Our responses are detailed below.

REVIEWERS' COMMENTS:

Reviewer #1:

Minor comments:

Fig. 1c is difficult to read. The text states that 20 out of 32 cells are silent in WC mode. I assume that these cells collapse in a single circle at the bottom of the bar. Although 12 cells should have non-zero firing rate, I distinguish only 5 non-zero markers in the graph. Could the author improve this panel's presentation, or its description in the legend?

This was a typographical error in the Results text. 26 (not 20) out of 32 GCs are silent in WC mode. One of the six non-zero values is 0.1 Hz so it looks very close to zero in the plot. We have updated the manuscript text and changed the legend to include this information.

Fig. 4c,d: how do you define the firing rate? It appears that you compute PSTH with a bin size of 50ms but this is not indicated in the text or legend.

Yes, this is how we defined firing rate in these plots. This information has been added to the legend.

Reviewer #2:

I remain somewhat unimpressed by the population coding implied by the piecewise linear fit in Fig. 5b. However, the data are displayed for all to interpret as they wish, and I don't believe the precise coding of the granule cells will be fundamental, as the downstream cells can presumably select desired inputs via learning and plasticity. Maybe the authors could start the y-axis of this plot at zero?

We have changed the y-axis to incorporate this recommendation.

Reviewer #3:

General Comments

The first general comment is the value-added to examining the relation between the responses and whisker position and set point. Set point is simply a low pass filtered version of whisker position. The two measures are highly correlated as shown in Figure 3a. One suspects that if one did a cross-correlation between whisker position and set point, the correlation would be quite high. Therefore, the neural responses in relation to these two measures will be similar as shown in Figure 3c and the finding that the neurons are correlated with both position and set point is not surprising. The authors would have to demonstrate that the two measures are really providing independent information on the behavior or should consider removing/downplaying.

We agree with the reviewer regarding the relationship between position and set point, but we feel that there is still added value by including both values in these plots. First, not all cells show such the close relationship between position and set point tuning (i.e. we also observe GCs modulated by phase, included in Supplementary Information). Second, inclusion of specific set point tuning helps to link the study with whisker kinematic research performed in other brain regions (e.g. Hill et al., 2011 Neuron in motor cortex), therefore we think these plots may enhance the impact of our data for researchers outside the cerebellar field.

The second general comment is the argument that the modulation depth for the GC responses is smaller for the input (EPSC) than for the output (firing rate) as shown in Figure 4h. However, whether this comparison is valid needs to be proven. To be valid the comparison needs to take into account the range and variability of the two measures, normalizing in a way to minimize the differences. For example, the range and ceiling effect of the two measures are very different. EPSC amplitudes will have a small range and variability with a ceiling after which the cell will fire, while firing rate in GCs can go from 0 to +200Hz. Given these differences would it even be possible for the depth of modulation of the EPSCs to be greater than for firing rate?

We believe that there is a misunderstanding here. We compare the modulation depth of EPSC rate (not amplitude as stated by the reviewer), with Spike rate. EPSC rates can be as high as 750-1000Hz (Rancz et al, 2007 Nature), so we are not seeing a physiological ceiling effect in our data. Also, we are voltage clamping cells when we measure EPSC rate so there is no chance that the cell can fire (and thus introduce an artificial experimental ceiling). Therefore we are convinced that the range of modulation is comparable for both measures.

Overall the measure we use is based on the ratio of the modulation range to its mean. Figure 4h (position and set point) and Supplementary Figure 3c (phase) both show that it is possible for the modulation depth of EPSC rate tuning to be greater than that of Spike rate tuning. The legend of Fig.4h states that the comparison is made between EPSC rate and Spike rate, and we hope this removes any further ambiguity.

Specific Comments

1) On line 129 should Fig. 1d be Fig. 1c?

Thank you, this has now been corrected.

2) In relation to Figure 4b, the text describes much higher bursting rates for GCs than the example selected. Why wasn't a more representative example of GC responses used? A similar question is for the spike rate in Figure 4c, that seems low when the burst rate is so high?

Very high firing rates mainly occur at preferred positions, and GCs are generally quite narrowly tuned. The data shown in Fig. 4b/4c is the response of a single GC to all movement epochs – therefore the average firing rates are significantly lower because they include GC responses at non-preferred positions. We show this data to allow direct and consistent comparison to be made between GC responses and MLI (and Purkinje cell) responses. However, we do still show representative firing at high rates – these are present on trials 4 and 5 in Fig 4b, and in Supplementary Figure 1.

3) On page 9, the text states there 45 INs were recorded, Should the statement on decreasing firing stating n=12/45 instead of n=12/35?

35 out of 45 cells showed significant modulation of firing rate with respect to movement. Of these 35 cells, 12 showed decreasing firing rates. This is stated at line 266.

4) Way was the reconstruction of the whisker trajectories done on the GC responses? It seems this would be very interesting comparison and help clarify the serial processing that is the theme of the study?

Most of our recordings were relatively short and GCs do not individually encode position via linear changes in firing rate. Therefore reconstruction attempts of single cells via linear transfer function are not appropriate. Also, our population is 'virtual', i.e. not acquired simultaneously, therefore the movement trajectories are not consistent and we cannot attempt a reconstruction on population data either. To perform such analysis, simultaneous population recordings from granule cells are required – we hope that this may be possible in the near future.